# Learning to Select Nodes in Branch and Bound with Sufficient Tree Representation

**Sijia Zhang, Shuli Zeng, Shaoang Li, Feng Wu** *, **Xiang-Yang Li** *
School of Computer Science and Technology, University of Science and Technology of China
{sxzsj,zengshuli0130,lishaoa}@mail.ustc.edu.cn
{wufeng02,xiangyangli}@ustc.edu.cn

## Abstract

Branch-and-bound methods are pivotal in solving Mixed Integer Linear Programming (MILP), where the challenge of node selection arises, necessitating the prioritization of different regions of the space for subsequent exploration. While machine learning techniques have been proposed to address this, two crucial problems concerning **(P1)** how to sufficiently extract features from the branch-and-bound tree, and **(P2)** how to assess the node quality comprehensively based on the features remain open. To tackle these challenges, we propose to tackle the node selection problem employing a novel Tripartite graph representation and Reinforcement learning with a Graph Neural Network model (TRGNN). The tripartite graph is theoretically proved to encompass sufficient information for tree representation in information theory. We learn node selection via reinforcement learning for learning delay rewards and give more comprehensive node metrics. Experiments show that TRGNN significantly improves the efficiency of solving MILPs compared to human-designed and learning-based node selection methods on both synthetic and large-scale real-world MILPs. Moreover, experiments demonstrate that TRGNN well generalizes to MILPs that are significantly larger than those seen during training.

## 1 Introduction

Mixed-integer linear programming (MILP) is a general optimization formulation for a wide range of important real-world applications (Zhang et al., 2023), such as industrial process scheduling (Floudas & Lin, 2005), resources allocation (Ren & Gao, 2010), and logistic operations (Paschos, 2014), etc. Modern MILP solvers (Gurobi Optimization, LLC, 2023; Bestuzheva et al., 2021) rely on the branch-and-bound (B&B) algorithm (Land & Doig, 1960), which recursively divides the search space into a tree, solving relaxations of the problem until an integral solution is found and proven optimal. Throughout this procedure, numerous decisions must be repeatedly made (Linderoth & Savelsbergh, 1999), including node selection, and branching variable selection, etc. These decisions often dramatically impact the overall performance (Kianfar, 2010). Traditionally, these would be made according to hard-coded expert heuristics implemented in solvers. To further improve the efficiency of MILP solvers, more and more attention is given to statistical learning approaches for replacing and improving upon those heuristics (Bengio et al., 2021).

Here, we focus on the **node selection** problem, which has a significant impact on the overall solver performance (Achterberg, 2007). Current machine learning-based node selection methods (He et al., 2014; Song et al., 2018; Yilmaz & Yorke-Smith, 2020; Labassi et al., 2022) predominantly extract features from a subset of candidate nodes, supplemented by manually extracted global tree features such as global upper and lower bounds. These methods primarily employ imitation learning to mimic an oracle which prioritizes a node if it contains the optimal solution.

However, the existing learning-based node selection methods (Labassi et al., 2022; Yilmaz & Yorke-Smith, 2020; He et al., 2014; Song et al., 2018) suffer from two limitations. Firstly, while individual MILP problems can be effectively represented using bipartite graphs (Gasse et al., 2019;

---

*Corresponding author.

Ding et al., 2020; Nair et al., 2020; Gupta et al., 2020; Paulus et al., 2022; Khalil et al., 2022; Shen et al., 2021; Labassi et al., 2022), node selection problem requires more than just representing a single MILP problem. It necessitates the representation of the entire branch-and-bound tree, which includes numerous nodes, each of which can be considered an individual MILP problem. However, representing all nodes in the tree as a bipartite graph is not feasible for large-scale problems due to the exponential increase in node count. For instance, the "Anonymous" dataset, rooted in a large-scale industrial context (Gasse et al., 2022), necessitates that SCIP handle upwards of 50,000 nodes. The previous work compares the nodes among the candidate nodes two by two to select the optimal node.

Table 1: Comparison between the expert strategies and SCIP in GISP($n \in [70, 80]$).

| Method | Time(s) | Wins | Nodes |
|---|---|---|---|
| SCIP | **7.54 ± 1.39** | **26/50** | 397.46 ± 2.59 |
| Expert | 7.82 ± 1.44 | 24/50 | **276.71 ± 2.82** |

They only represent the two currently selected nodes to be compared as a bipartite graph, supplemented by manually extracted global features, such as global upper and lower bounds and the depth of nodes. However, these manually extracted features do not fully capture the solving process of the branch-and-bound tree. Secondly, the recent work (Labassi et al., 2022; Yilmaz & Yorke-Smith, 2020; He et al., 2014; Song et al., 2018), which is based on the imitation learning approach, is limited by the performance of the expert itself. However, the oracle sometimes fails to outperform SCIP, as observed in datasets like GISP (Colombi et al., 2017), as shown in Table 1. If the expert does not beat a baseline, imitating it is unlikely to bring gains.

To overcome the aforementioned challenges, we propose to tackle the node selection problem employing a **T**ripartite graph representation and **R**einforcement learning with a **G**raph **N**eural **N**etwork model (TRGNN). Firstly, we propose a tripartite graph to represent the branch-and-bound tree, which is theoretically proved to be sufficient in information theory. Secondly, we introduce a reinforcement learning framework for learning delay rewards in the branch-and-bound process and give more comprehensive node metrics. These metrics include the global gap, the node potential and the overhead of path switching, which is the transition between the current node and the subsequent selected one.

We evaluate our approach on six NP-hard MILP problem benchmarks, which consist of three classical synthetic MILP problems and three real-world MILP problems from diverse application areas. Experiments show that TRGNN significantly outperforms human-designed and learning-based node selection methods in terms of solving time. We achieve up to 31.44% reduction on synthetic instances and up to 59.42% reduction on real-world instances. Notably, TRGNN is the only known learning-based node selection method to surpass the default SCIP across all of the six datasets in terms of solving time. Moreover, experiments show that TRGNN can well generalize to MILPs that are significantly larger than those seen during training.

## 2 RELATED WORK

He et al. (2014) were the first to explore node selection heuristics in branch-and-bound by applying imitation learning. They trained a support vector machine (SVM) to replicate the decisions of an oracle that selects nodes along the optimal solution path. Their model leveraged hand-crafted features, including node-specific attributes and global metrics like the number of solutions found and the current global bounds. However, their approach primarily focused on combining node selection with a learned pruning model, which aimed to cut off unpromising branches in the branch-and-bound tree. Their strategy was more akin to a primal heuristic, emphasizing the quick discovery of high-quality feasible solutions rather than guaranteeing optimality. They reported improvements in the optimality gap against SCIP under a time limit and Gurobi under a node limit on four benchmarks.

Song et al. (2018) proposed a RankNet model combined with a novel retrospective imitation learning method. In contrast to He et al. (2014), they focused solely on node selection without an additional pruning operator. They retrospectively correct the node selection trajectory into a shortest path to the best solution found. When the solver runs until optimality, the trajectory mirrors those generated by a diving oracle. They generated these trajectories using Gurobi and trained the model with the DAGGER (Ross et al., 2011) and SMILe (Ross & Bagnell, 2010) imitation learning algorithms. Their method demonstrated significant improvements in the optimality gap under a node limit on path planning integer programs. However, they reported less favorable results on

more challenging benchmarks, such as combinatorial auctions, indicating limitations in the model's generalizability to more complex problems.

Yilmaz & Yorke-Smith (2020) extended the work of He et al. (2014) by proposing a neural network-based node selection operator for branch-and-bound. Their method can be viewed as a refinement of Song et al. (2018)'s neural network node selector, with the key difference being that it only evaluates children of the current node and defaults to depth-first search otherwise. Using the state encoding from Gasse et al. (2019), they trained their model to mimic an oracle that prioritizes nodes leading to one of the top k solutions, essentially generalizing He et al. (2014)'s oracle. On three benchmarks, they reported improvements in both time and number of nodes, and in some cases, reduced node counts compared to SCIP. However, their approach still lagged behind SCIP in terms of solving time, showing limitations in overall efficiency despite the node count improvements.

Inspired by the work of Gasse et al. (2019) for variable selection in branch-and-bound, Labassi et al. (2022) considered a node selection strategy that compared nodes in the candidate queue pair by pair to identify the optimal node. They represented only the two currently selected nodes as bipartite graphs, supplemented by manually extracted global features such as the global upper and lower bounds and the nodes' depth. On three benchmarks, they reported reductions in the node counts of the branch-and-bound tree and demonstrated time improvements over SCIP on one dataset. However, like earlier efforts, their method underperformed on other datasets in terms of solving time. Notably, even the oracle failed to outperform SCIP's default rule on certain datasets. Consequently, it is unsurprising that no other machine learning approach was able to surpass SCIP in these cases.

Some works have sought to provide a full tree representation. Scavuzzo et al. (2022) address the variable selection problem by proposing tree Markov Decision Processes that offers a more suitable framework for learning to branch. On the other hand, Mattick & Mutschler (2023) focus on node selection, addressing the complexity of batching variable and constraint structures. Instead of creating $k$ vertices for $k$ variables, they represent variable values as a distribution with 10 buckets. This method allows for scaling to much larger instances than typically possible, although it sacrifices some detail in representation, thus leaving room for optimization and fine-tuning.

## 3 PRELIMINARIES

### 3.1 MIXED INTEGER LINEAR PROGRAMMING

A general MILP problem is defined by a set of decision variables, where a subset or all variables are required to be integers. The objective is to maximize a linear function under a series of linear constraints, as formulated below:

$$\max \boldsymbol{c}^\top \boldsymbol{x}, \quad \text{s.t.} \quad \boldsymbol{A}\boldsymbol{x} \geq \boldsymbol{b}, \boldsymbol{x} \in \mathbb{R}^l, \quad x_j \in \mathbb{Z}, \forall j \in \mathcal{I}, \tag{1}$$

For simplicity, we assume that the objective of our MILP problems is to seek the maximum value.

An MILP problem can be effectively represented as a weighted bipartite graph $G = (V \cup C, E)$ (Nair et al., 2020; Gasse et al., 2019). Each vertex in $V$ corresponds to a variable of the MILP, and each vertex in $C$ represents a constraint. An edge $(v_i, c_j)$ connects a variable vertex $v_i$ with a constraint vertex $c_j$ if the variable is involved in the constraint. The edge set $E \in \mathbb{R}^{l \times m \times e}$ represents the edge features, where $l$ and $m$ denote the number of variables and constraints, respectively, and $e$ indicates the dimension of the edge attributes.

### 3.2 BRANCH AND BOUND

The branch and bound (Land & Doig, 1960) algorithm can be described as follows. At every node, the linear program (LP) relaxation is solved, where all variables are treated as continuous. If the LP relaxation solution $x_{N_s}$ of the selected node $N_s$ violates the original integrality constraints, the problem "branches" into two sub-MILPs (child nodes) by adding constraints that force a fractional variable to round up or down. Specifically, the leaf node is added with constraints $x_i \leq \lfloor (x_{N_s})_i \rfloor$ and $x_i \geq \lceil (x_{N_s})_i \rceil$, respectively, where $x_i$ denotes the $i$-th variable, $(x_{N_s})_i$ denotes the $i$-th variable of the LP relaxation solution $x_{N_s}$. If the solution $x_{N_s}$ is integer (and feasible for the original MILP as per Equation 1), and its objective value surpasses the current best integer feasible solution, it is designated as the new global lower bound. If the objective value $f(x_{N_s})$ (i.e., the node upper bound) is lower than the global lower bound, or if the LP problem is infeasible, the node is pruned.

Figure 1: Example of tripartite graph representation. The root node (red) is conceptualized as a bipartite graph, consisting of variable and constraint vertices, while the leaf nodes (grey) embody sets of added branching constraints. The features of the edges, which connect the variable vertices to the node constraint vertices, delineate the constraint space of the leaf nodes.

## 3.3 INFORMATION THEORY

In information theory (Reza, 1994), mutual information $I(X;Y)$ is the amount of uncertainty in $X$ due to the knowledge of $Y$. Mathematically, mutual information is defined as

$$I(X;Y) = \sum_{x,y} p(x,y) \log(\frac{p(x,y)}{p(x)p(y)}) \tag{2}$$

where $p(x,y)$ is the joint probability distribution function of $X$ and $Y$, and $p(x)$ and $p(y)$ are the marginal probability distribution functions for $X$ and $Y$. We can also say

$$I(X;Y) = H(X) - H(X|Y) \tag{3}$$

where $H(X)$ is the marginal entropy, $H(X|Y)$ is the conditional entropy. If $H(X)$ represents the measure of uncertainty about a random variable, then $H(X|Y)$ measures what Y does not say about X. This is the amount of uncertainty in $X$ after knowing $Y$ and this substantiates the intuitive meaning of mutual information as the amount of information that knowing either variable provides about the other.

## 4 TRIPARTITE GRAPH REPRESENTATION OF BRANCH AND BOUND TREE

While a sufficient representation of every node within a branch-and-bound tree $T$ would theoretically capture the entire scope of tree information, such an approach is not feasible for large-scale problems due to the exponential increase in node count. For instance, the Anonymous dataset (Gasse et al., 2022), rooted in a large-scale industrial context, necessitates that SCIP handle upwards of 50,000 nodes. We define the complete tree feature vector, $F = \bigoplus_{N \in T} J_N$, by aggregating features $J_N \in \mathcal{J}$ for each node $N$. This raises a critical question: Which node features within $F$ are essential and should be selectively represented to effectively model the tree? Addressing this, we introduce a tripartite graph structure for tree representation in Section 4.1.

In order to judge whether a tree representation effectively can capture all relevant node features necessary for node quality assessment, we first need to define what a sufficient tree representation is. Let $\phi(\cdot)$ denote a feature extraction function, which depends on the specific methods employed for tree representation. The feature vector, $\phi(F)$, is derived from the tree complete feature $F$. Shannon's information theory provides a suitable formalism for quantifying the above concepts. The information about the quality vector $Q$ contained in the feature vector $\phi(F)$ is then $I(Q;\phi(F))$. As a consequence of the data processing inequality, $I(Q;\phi(F)) \leq I(Q;F)$. A feature vector $\phi(F)$ is defined to be sufficient if the inequality above is an equality.

**Definition 4.1** (Definition of Sufficient Tree Representation). *A tree feature representation $\phi(F)$ is considered sufficient if:*

$$I(Q;\phi(F)) = I(Q;F)$$

In Section 4.2, we theoretically demonstrate that the tree information captured within this graph is sufficient for node selection.

## 4.1 TREE REPRESENTATION

In this section, we introduce a tripartite graph for tree representation. An advantage of our representation is that it is theoretically proved to be sufficient in Section 4.2. This theoretical suffi-

ciency ensures that our representation can comprehensively capture the necessary features across the branch-and-bound tree for effective node quality assessment. Additionally, to enhance practical applicability in machine learning, we tailor the representation to facilitate feature learning. Instead of directly representing each node with its variables and constraints, which would redundantly present at the root node, we focus on depicting the unique branch constraints that distinguish each node. This approach emphasizes the subtle, often overlooked differences among nodes, thereby facilitating machine learning algorithms to more effectively learn and discriminate node features.

We represent the branch-and-bound tree as a weighted tripartite graph $G = (V \cup C \cup LN, E^C \cup E^{LN})$. The vertex set $V \cup C \cup LN$ is divided into three subsets: the variable vertex set $V$ with $|V| = l$, the constraint vertex set $C$ with $|C| = m$ and the leaf node vertex set $LN$ with $|LN| = n$. The vertex sets $V$, $C$ and $LN$ are mutually exclusive: $V \cap C = V \cap LN = C \cap LN = \emptyset$. We denote the variable vertices as $V = \{v_1, v_2, \ldots, v_l\}$, constraint vertices as $C = \{c_1, c_2, \ldots, c_m\}$ and leaf node vertices $LN = \{ln_1, ln_2, \ldots, ln_n\}$. The leaf node vertices represent candidate leaf nodes for selection strategies within the branch-and-bound tree. After a node is selected, the branch-and-bound algorithm divides it into two new leaf nodes, adding constraints of the form $\{x_i \leq z\}$ or $\{x_i \geq z\}$, where $x_i$ is the variable chosen for branching and $z \in \mathbb{Z}$. The constraint edges $E^C$ connect variable vertices $V$ to constraint vertices $C$ and the branching edges $E^B$ connect variable vertices $V$ to leaf node vertices $LN$, indicating branching decisions. The edges are denoted as $E_{i,j}^C = (v_i, c_j)$ and $E_{j,k}^B = (c_j, ln_k)$, with $|E^C| = e_1, |E^B| = e_2$, for $i \in \{1, 2, \ldots, l\}, j \in \{1, 2, \ldots, m\}, k \in \{1, 2, \ldots, n\}$. We denote the collection of all such weighted tripartite graphs $G = (V \cup C \cup LN, E^C \cup E^{LN})$ with $|V| = l, |C| = m$ and $|LN| = n$ as $\mathcal{G}_{l,m,n}$.

Each vertex in the tripartite graph is associated with a feature vector: vertices in $V$ (variables), $C$ (constraints), and $LN$ (leaf nodes) hold features $h_i^V \in \mathcal{H}^V$, $h_j^C \in \mathcal{H}^C$, and $h_k^{LN} \in \mathcal{H}^{LN}$, respectively. We define the feature spaces for the sets of all variable vertices, constraint vertices, and leaf node vertices as $\mathcal{H}_l^V := (\mathcal{H}^V)^l$, $\mathcal{H}_m^C := (\mathcal{H}^C)^m$, and $\mathcal{H}_n^{LN} := (\mathcal{H}^{LN})^n$. Edges in the graph are imbued with specific attributes that detail the interactions between vertices. For constraint edges that connect constraint vertices $c_i$ to variable vertices $v_j$, each edge feature $f_{i,j}^C \in \mathcal{F}^C$ is one-dimensional and directly represents the coefficient $\boldsymbol{A}_{ij}$ of variable $x_j$ in constraint $c_i$. Branching edges connect variable vertices $v_j$ to leaf node vertices $ln_k$, and these edges carry two-dimensional features that specify the branching conditions for the variable $x_j$ at the leaf node. Each branching edge feature encapsulates $x_j \geq a$ and $x_j \leq b$, represented as $f_{j,k}^B = (a, b) \in \mathcal{F}^B$. We define $\mathcal{F}_{e_1}^C := (\mathcal{F}^C)^{e_1}$ and $\mathcal{F}_{e_2}^B := (\mathcal{F}^B)^{e_2}$. The exact features of the graph are described in Appendix C.

We extract an MILP solving search tree's features via a tripartite graph $\phi_t(F) = (V \cup C \cup LN, E^C \cup E^B) \in \mathcal{G}_{l,m,n} \times \mathcal{H}_l^V \times \mathcal{H}_m^C \times \mathcal{H}_n^{LN} \times \mathcal{F}_{e_1}^C \times \mathcal{F}_{e_2}^B$, where $F$ denotes the aggregate of node features. In Figure 1, we illustrate the process of converting a branch-and-bound tree from an MILP instance into a tripartite graph. The conversion begins with the root node, where variables and constraints are represented as "variable vertices" and "constraint vertices", respectively. As the branching process progresses, each step adds new constraints to the leaf nodes, encapsulated as "leaf node vertices" in the graph. These additional constraints differentiate each leaf node's MILP problem from the root node's problem. Edges connect these leaf node vertices to variable vertices, forming a path that represents the sequence of branching decisions made from the root node to each leaf node. For example, in Figure 1, the candidate leaf node in the lower left undergoes two branching steps, adding constraints $y \leq 1$ and $x \leq 1$, represented by edges $e_1$ and $e_3$, respectively.

## 4.2 THEORETICAL PROOF OF SUFFICIENCY

In this section, we prove our main theorems and present a sketch of our proof lines. The full proof lines are presented in the Appendix A. We first present with the following theorem that the tripartite graph is sufficient to represent an MILP search tree.

**Theorem 4.1.** *Given the node $N_i$ with features $J_{N_i}$, the tripartite graph representation $\phi_t(F)$ satisfies: $I(Q; \phi_t(F)) = I(Q; F)$, where $F = \bigoplus_{i=0}^{m+n} J_{N_i}$ and $Q$ denotes the quality vector.*

This theorem asserts that the information uncertainty in $Q$, when knowing the tripartite graph representation $\phi_t(F)$, is equivalent to that when knowing the features of all nodes in the tree $F$.

Then, we analyze the relationship between the features of child nodes and their parent node:

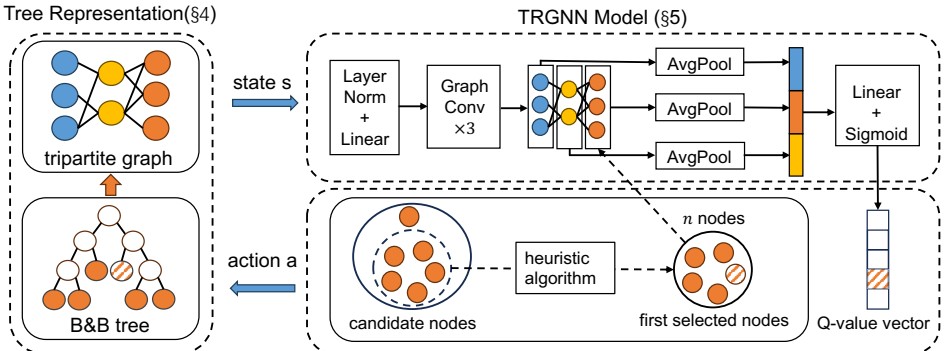

Figure 2: Illustration of our proposed RL framework for learning node selection policies. In this framework, the search tree is represented as a tripartite graph (see Section 4), serving as the environment, and the TRGNN model acts as the agent(see Section 5). We first apply the heuristic algorithm "BestEstimate" to pre-select $n$ candidate nodes and incorporate their estimate values into the node features. Subsequently, the TRGNN model processes these features and outputs a Q-value vector, from which the nodes with the highest Q-values are selected as the final choices.

**Theorem 4.2.** *Given a node $N_p$ and its two child nodes $N_l$ and $N_r$, it holds that $I(Q; J_{N_p} \oplus J_{N_l} \oplus J_{N_r}) = I(Q; J_{N_l} \oplus J_{N_r})$, where $J_N$ denotes the extracted feature of the node $N$.*

This theorem illustrates that the information provided by the features of the parent node can be completely represented by the features of its child nodes. The proof proceeds as follows: We start with the obvious inequality: $I(Q; J_{N_p} \oplus J_{N_l} \oplus J_{N_r}) \geq I(Q; J_{N_l} \oplus J_{N_r})$ according to the data processing inequality. To establish the reverse inequality, we analyze the node features, encompassing both MILP characteristics, denoted by $J_m$, including the variables, constraints and objective function, and the global attributes within the tree, denoted by $J_g$. Thus, we define: $J_N = (J_m)_N \oplus (J_g)_N$. We demonstrate that functions $f_m$ and $f_g$ exist such that $(J_m)_{N_p} = f_m((J_m)_{N_l} \oplus (J_m)_{N_r})$ and $(J_g)_{N_p} = f_g((J_g)_{N_l} \oplus (J_g)_{N_r})$. This relationship establishes a Markov chain: $Q \rightarrow J_{N_l} \oplus J_{N_r} \rightarrow J_{N_p} \oplus J_{N_l} \oplus J_{N_r}$, allowing the use of the data processing inequality to confirm $I(Q; J_{N_p} \oplus J_{N_l} \oplus J_{N_r}) \leq I(Q; J_{N_l} \oplus J_{N_r})$.

By invoking Theorem 4.2 recursively, we illustrate that the features of all leaf nodes, denoted by $L$, encapsulate the necessary information to represent the whole tree, thereby establishing that $I(Q; L) = I(Q; F)$. This finding is pivotal for proving Theorem 4.1.

In proving Theorem 4.1, it is essential to validate that the information flow $Q \rightarrow \phi_t(F) \rightarrow L$ holds. The tripartite graph representation $\phi_t(F)$ includes the root node's MILP features $(J_m)_{N_0}$ and each leaf node's branching constraints $B_N$ and global features $(J_g)_N$, ensures that from $(J_m)_N$ and $B_N$, the MILP features $(J_m)_N$ of the leaf node $N$ can be deduced. Employing the data processing inequality again, we establish the sufficiency of $\phi_t(F)$ as a representation, proving Theorem 4.1.

## 5 LEARNING NODE SELECTION VIA TRIPARTITE GRAPH

Building upon the tripartite graph framework detailed in Section 4 for extracting observable features from the branch-and-bound tree **(P1)**, this section delves into **(P2)**: assessing node quality and selecting a node based on the tripartite graph representation.

The challenges in node selection are twofold. Firstly, the branch-and-bound process involves sequential decision-making, where each choice impacts subsequent decisions. This creates a scenario of delayed rewards, where the consequences of early decisions may not become apparent until later stages. Secondly, there is a distinct lack of comprehensive metrics for evaluating node quality.

To address these issues, we frame the node selection problem within a Markov Decision Process (MDP) framework and leverage Reinforcement Learning (RL) to learn node selection policies, as detailed in Section 5.1. Following this, we introduce three node quality metrics that assess the impact of nodes in the solving process, further discussed in Section 5.2.

## 5.1 REINFORCEMENT LEARNING FORMULATION

We formulate an MILP solver as the environment and the RL model as the agent. We consider an MDP defined by the tuple $(\mathcal{S}, \mathcal{A}, r, \pi)$. Specifically, we specify the state space $\mathcal{S}$, the action space $\mathcal{A}$, the reward function $r : \mathcal{S} \times \mathcal{A} \to \mathbb{R}$, the transition function $\pi$, and the terminal state in the following. **(1) The state space $\mathcal{S}$.** As delineated in Section 4, the core information for node selection is represented by a tripartite graph. **(2) The action space $\mathcal{A}$.** The action space is intended to include all nodes that are potentially selectable. However, the dynamic nature of the selection process means the number of selectable nodes is subject to change due to the addition of newly expanded nodes and the removal of pruned nodes. To address this variability, we employ the heuristic node selection algorithm called $Estimate$ in modern solvers to pre-select nodes, choosing the top $n$ nodes, where $n$ is a predetermined value, to form a set of node candidates. If the initial set of candidates is less than $n$, placeholders are used to fill the remaining slots, ensuring a consistent set size. We define the action space as this set of node candidates with a size of $n$. **(3) The reward function $r$.** The reward function, as discussed in Section 5.2, encompasses the gap update reward, the optimal solution path reward, and the path switching penalty. **(4) The transition function $\pi$.** The transition function maps the current state $s$ and the action $a$ to the next state $s'$, representing the ensuing search tree post the expansion of node $a$. **(5) The terminal state.** The process reaches a terminal state when the gap attains zero or no remaining candidate nodes are in the set. Within this framework, the trajectory probability $\tau = (s_0, \ldots, s_T)$ depends on the node selection policy $\pi$ and the other solver components, formulated as $p_\pi(\tau) = p(s_0) \prod_{t=0}^{T-1} \sum_{a \in \mathcal{A}} \pi(a|s_t) p(s_{t+1}|s_t, a)$.

**TRGNN Model.** Figure 2 delineates the architecture of our proposed model. The features of the variable, constraint, and leaf node vertices on the tripartite graph undergo an initial transformation via a 32-dimensional embedding layer. This layer is pivotal for normalizing and refining the input features before they traverse through three subsequent graph convolutional layers, each with dimensions 8, 4, and 4, and each utilizing a ReLU activation function to capture complex, non-linear relationships. Post convolution, the refined representations of the variable, constraint, and node constraint vectors are separately averaged, reducing dimensionality and mitigating overfitting risks. These averaged representations are then amalgamated with the global node features, creating a comprehensive feature vector encapsulating both localized and global information. This amalgamated feature vector is then processed through a linear layer and a sigmoid activation layer, culminating in a Q-value vector of predetermined dimension $n$. This vector quantitatively represents the value of each node in the pre-selected candidate node set, serving as a decisive metric for action selection. The action is determined by selecting the node corresponding to the maximum Q-value, directing the model's focus towards the most promising regions of the search space.

A significant advantage of our model is higher learning efficiency for different node features. Experiments demonstrate that TRGNN requires far fewer training examples compared to conventional imitation learning models (Labassi et al., 2022), thus conserving computational resources. For instance, with the GISP datasets, our model needs only 250 instances for training, whereas traditional imitation learning methods may require up to 1000 instances.

## 5.2 REWARD FUNCTION

The reward function $R$ for the selected node $N$ is defined as:

$$R(N) = w_1 R_g(N) + w_2 R_o(N) - w_3 R_s(N) \tag{4}$$

where $R_g$ represents the updates to the global gap, $R_o$ represents the potential of a node to lead to an optimal solution, and $R_s$ signifies the normalized reward for path-switching steps. Due to space constraints, the normalizing methods and parameters are detailed in the Appendix.

In branch-and-bound algorithms, node selection is crucial for optimizing the solving process. The overarching goal is to efficiently converge to the optimal solution by narrowing the primal and dual bounds. Each node is critically evaluated for its potential to: (1) update the global lower bound, finding an improved feasible solution, (2) influence the global upper bound, reducing the gap between the current best known solution and the optimal solution, or (3) generate promising descendant nodes that might yield feasible solutions.

The updates to the global lower and upper bounds are quantitatively assessed by changes in the global gap. We denote the optimal solution reward $R_o$, a methodology endorsed by prior re-

search (He et al., 2014; Yilmaz & Yorke-Smith, 2020; Labassi et al., 2022).

$$R_o(N) = \begin{cases} 1 & \text{if } LB_N \leq x^* \leq UB_N, \\ 0 & \text{otherwise.} \end{cases} \tag{5}$$

where $LB_N$ and $UB_N$ denotes the local lower and upper bound of node $N$, $x^*$ denotes the optimal solution. Each node defines a unique segment of the search space; If the optimal solution resides within the current node's domain, the reward $R_o$ is 1. If the optimal solution is not within the domain of the current node, the reward for this component is zero. This method enables the model to learn from node features to discern which nodes are likely to lead to potentially promising solutions.

However, one often-overlooked aspect of node selection is the time taken to transition from the last selected node to the newly chosen one. This transition time can significantly impact overall performance. For instance, in the MaxSAT dataset, where $n$ ranges from 70 to 80, the path switching phase consumes, on average, 5.2% of the total solving time. In the branch-and-bound process, solvers like SCIP (Bestuzheva et al., 2021) navigate the path from the last selected node to the newly chosen one, known as path switching.

Figure 3 illustrates how each node traces back through the tree to find a common ancestor. Although this process might seem straightforward, it introduces significant computational overhead. We employ the "Eventhdlr" from PySCIPOpt (Maher et al., 2016). We utilize the "PY_SCIP_EVENTTYPE" states "NODE-SOLVED" and "NODEFOCUSED" to track the time from when one node is completely solved ("NODESOLVED") to when the solver shifts its focus to another node ("NODEFO-CUSED"). Since the path switch time is positively correlated with the number of path switch steps, we incorporate the steps required for a path switch as a component of the reward func-

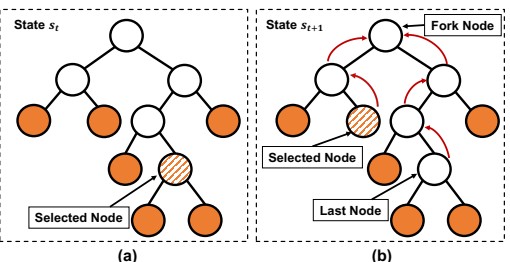

Figure 3: Path switching process. Figure (a) depicts the current state $s_t$ with node $N_t$ selected and figure (b) shows the subsequent state $s_{t+1}$ with node $N_{t+1}$ chosen.

tion in our reinforcement learning model. We calculate the number of steps between nodes by adding the number of steps each node takes to reach their nearest common ancestor.

## 6 EXPERIMENTS

Table 2: Policy evaluation on the synthetic and real-world datasets. The best performance is marked in bold. We report the 1-shifted geometric mean (standard deviation) of the Time.

| Methods | FCMCNF | | | MaxSAT | | | GISP | | |
|---|---|---|---|---|---|---|---|---|---|
| | Time(s) | wins | Im(%) | Time(s) | wins | Im(%) | Time(s) | wins | Im(%) |
| SCIP | 21.15 ± 2.42 | 20/100 | 0.00 | 8.22 ± 1.90 | 23/100 | 0.00 | 3.88 ± 1.32 | 27/100 | 0.00 |
| SVM | 22.17 ± 2.44 | 14/100 | -4.82 | 8.15 ± 1.76 | 8/100 | +0.85 | 4.98 ± 1.78 | 3/100 | -28.35 |
| RankNet | 22.39 ± 2.24 | 14/100 | -5.86 | 8.29 ± 1.83 | 9/100 | -0.85 | 5.03 ± 1.81 | 2/100 | -29.64 |
| GNN | 19.63 ± 2.13 | 18/100 | +7.19 | 8.05 ± 1.98 | 17/100 | +2.07 | 4.19 ±1.79 | 13/100 | -7.99 |
| TRGNN (Ours) | **14.50 ± 2.21** | **34/100** | **+31.44** | **7.60 ± 1.86** | **43/100** | **+7.54** | **3.34 ± 1.37** | **55/100** | **+13.92** |

| Methods | MIK | | | CORLAT | | | Anonymous | | |
|---|---|---|---|---|---|---|---|---|---|
| | Time(s) | wins | Im(%) | Time(s) | wins | Im(%) | Time(s) | wins | Im(%) |
| SCIP | 29.45 ± 1.76 | 0/18 | 0.00 | 6.81 ± 5.00 | 6/50 | 0.00 | 215.28 ± 37.90 | 0/20 | 0.00 |
| SVM | 39.60 ± 1.79 | 0/18 | -34.46 | 4.78 ± 4.42 | 4/50 | +29.81 | 1966.14 ± 25.06 | 0/20 | -8.13 |
| RankNet | 48.00 ± 1.95 | 0/18 | -62.99 | 4.52 ± 3.81 | 5/50 | +33.63 | 1840.65 ± 48.86 | 0/20 | -7.55 |
| GNN | 40.08 ± 1.69 | 2/18 | -36.99 | 4.71 ± 3.90 | 8/50 | +30.84 | 271.40 ± 15.38 | 0/20 | -26.07 |
| TRGNN (Ours) | **11.95 ± 1.45** | **16/18** | **+59.42** | **4.23 ± 4.10** | **19/50** | **+37.89** | **105.68 ± 20.87** | **20/20** | **+50.91** |

Our experiments have three main parts: **Experiment (1)** Evaluate our approach on three classical MILP problems and three challenging MILP problem benchmarks from diverse application areas. **Experiment (2)** Perform carefully designed ablation studies to provide further insight into TRGNN.

**Experiment (3)** Test whether TRGNN can generalize to instances significantlly larger than those seen during training. The codes are modified from Labassi et al. (2022).

## 6.1 SETUP

**Benchmarks.** We evaluate our approach on six NP-hard MILP problem benchmarks, which consist of three classical synthetic MILP problems and three real-world MILP problems from diverse application areas. (1) Synthetic datasets comprise three widely used synthetic MILP problem benchmarks: Fixed Charge Multicommodity Network Flow (FCMCNF) (Hewitt et al., 2010), Maximum Satisfiability (MAXSAT) (Ansótegui & Gabàs, 2017) and Generalized Independent Set (GISP) (Colombi et al., 2017). We artificially generate instances following Béjar et al. (2009); Chmiela et al. (2021); Labassi et al. (2022). (2) Real-world datasets comprise MIK (Atamtürk, 2003), CORLAT (Gomes et al., 2008), and the Anonymous problem, inspired by a large-scale industrial application (Gasse et al., 2022). Details of datasets are provided in the Appendix D.1. Throughout all experiments, we use SCIP 8.0.4 (Bestuzheva et al., 2021) as the backend solver, which is the state-of-the art open source solver, and is widely used in research of machine learning for combinatorial optimization (Chmiela et al., 2021; Gasse et al., 2019; Turner et al., 2022; Wang et al., 2023). We keep all the other SCIP parameters to default and emphasize that all of the SCIP solver's advanced features, such as presolve and heuristics, are open.

**Baselines.** We compare against the state-of-the-art best estimate node selection rule (Bénichou et al., 1971; Forrest et al., 1974). This is the default method in SCIP (Bestuzheva et al., 2021). In addition, we compare against three machine learning approaches: the Support Vector Machine (SVM) approach (He et al., 2014), the RankNet feedforward neural network approach (Song et al., 2018), and the approach based on Graph Neural Networks (GNN) (Labassi et al., 2022).

**Training.** We generate extensive synthetic datasets, consisting of 10,000 training samples and 1,000 test samples. From these, we randomly select 1,000 samples for the SVM, RankNet, and GNN models training and 100 samples for testing within each problem. Additionally, we generate 1,000 larger-scale datasets, randomly selecting 50 samples from each to evaluate the transfer capability of our method. For the real-world dataset, each problem set is split into training and test sets with 80% and 20% of the instances. Specifically, we train model with 1,000 samples for Corlat, 72 samples for MIK and 65 samples for Anonymous dataset. The model implemented in PyTorch (Paszke et al., 2019) and optimized using Adam (Kingma & Ba, 2014) with training batch size of 16. The training process is conducted on a single machine that contains eight GPU devices(NVIDIA GeForce RTX 4090) and two AMD EPYC 7763 CPUs.

We observe that our model performs remarkably well even when trained with fewer samples, achieving results comparable to those obtained with larger datasets. This demonstrates its efficiency and effectiveness in utilizing limited data. The ability to maintain such performance with a reduced dataset size is particularly valuable, given the significant time and computational resources required to collect training samples by solving multiple MILP instances. For instance, collecting 90 samples for the Anonymous problem alone took over 13.89 hours. In the Appendix D.5, we provide a comparison between our TRGNN model and a GNN baseline trained with the same limited dataset, referred to as GNN_light.

**Evaluation Metrics.** We employ two widely recognized evaluation metrics: (1) Time: running times in seconds (lower is better) and (2) Wins: number of times each node selection strategy resulted in the fastest solving time, over total number of solved instances. We limit the solution time for all problems to a maximum of 3600 seconds. Any time exceeding this limit is recorded as 3600 seconds. We assess node selection methods in terms of the 1-shifted geometric mean over the instances, accompanied by the geometric standard deviation. Furthermore, to evaluate different node selection methods compared to the default SCIP, we propose an (3) Improvement metric. Specifically, we define the metric by $Im = \frac{T(SCIP)-T(M)}{T(SCIP)}$, where $T(SCIP)$ represents the solving time of SCIP, and $T(M)$ represents the solving time of the compared method. Due to the space limits, we report the size of the branch-and-bound tree (Nodes) in the Appendix D.4.

## 6.2 COMPARATIVE EXPERIMENT

For each problem, machine learning models are trained on instances of the same size as the test instances. The results in Table 2 suggest TRGNN significantly outperforms all baseline methods.

Importantly, it is the only known learning-based node selection method to consistently outperform the default SCIP in terms of solving time across all six datasets. Compared to SCIP, TRGNN demonstrates notable efficiency improvements across all tested problems, being 31.44% faster in FCMCNF, 7.54% in MaxSAT, 13.92% in GISP, 59.42% in MIK, 37.89% in CORLAT, 50.91% in Anonymous.

## 6.3 ABLATION STUDY

We present an ablation study conducted to evaluate the contribution of different metrics in Sec. 5. We report the performance of TRGNN, TRGNN-1 and TRGNN-2 with different reward strategies in Table 3 on FCMCNF, GISP, MIK, and Anonymous. TRGNN-1 utilizes our RL framework with a reward function based on the optimal solution, as established in prior research (Labassi et al., 2022).

TRGNN-2 incorporates gap change into the node metrics, while TRGNN, our full model, additionally accounts for path switching overhead. The results in Table 3 reveal that both TRGNN-1 and TRGNN-2 outperform SCIP on GISP, MIK, and Anonymous datasets, highlighting the benefits of integrating the reinforcement learning model with our tripartite graph representation. Furthermore, TRGNN significantly surpasses TRGNN-1, TRGNN-2, and other baselines across all four datasets. The

Table 3: Comparison between TRGNN, TRGNN with two metrics (TRGNN-2) and TRGNN with one metric (TRGNN-1).

| Methods | FCMCNF | GISP | MIK | Anonymous |
|---|---|---|---|---|
| SCIP | 21.15 s | 3.88 s | 29.45 s | 215.28 s |
| TRGNN-1 | 23.28 s | 3.48 s | 13.27 s | 183.62 s |
| TRGNN-2 | 22.97 s | 3.36 s | 12.08 s | 124.05 s |
| TRGNN(Ours) | **14.50 s** | **3.34 s** | **11.95 s** | **105.68 s** |

results demonstrate that both of the gap update and path switching overhead is important for efficient exploration in complex tasks.

## 6.4 GENERALIZATION

We evaluate the ability of TRGNN to generalize across larger sizes of MILPs. We evaluate the generalization ability on FCMCNF, MaxSAT and GISP datasets, as we can artificially generate the larger transfer instances.

Table 4: The generalization ability of TRGNN. The best performance is marked in bold. We report the 1-shifted geometric mean (standard deviation) of the Time.

| Methods | FCMCNF | | | MaxSAT | | | GISP | | |
|---|---|---|---|---|---|---|---|---|---|
| | Time(s) | wins | Im(%) | Time(s) | wins | Im(%) | Time(s) | wins | Im(%) |
| SCIP | 1474.60 ± 4.19 | 3/50 | 0.00 | 76.12 ± 1.83 | 4/50 | 0.00 | 427.29 ± 1.65 | 0/50 | 0.00 |
| SVM | 1857.29 ± 3.99 | 0/50 | -25.95 | 84.78 ± 2.66 | 1/50 | -11.37 | 557.52 ± 2.39 | 0/50 | -30.48 |
| RankNet | 1560.22 ± 4.18 | 7/50 | -5.80 | 101.45 ± 2.15 | 0/50 | -33.27 | 701.62 ± 2.02 | 0/50 | -64.20 |
| GNN | 1503.83 ± 3.90 | 11/50 | -1.98 | 69.44 ± 1.57 | 19/50 | +8.78 | 534.72 ± 1.90 | 0/50 | -25.14 |
| TRGNN (Ours) | **1243.39 ± 3.64** | **29/50** | **+15.68** | **48.20 ± 1.55** | **26/50** | **+36.67** | **339.66 ± 1.85** | **50/50** | **+20.51** |

The results in Table 4 indicate that TRGNN significantly surpasses the baselines in terms of both solving time and number of wins, demonstrating its superior generalization ability. Notably, TRGNN demonstrates an even more pronounced improvement in solving time on larger datasets compared to smaller ones, underscoring the importance of effective node selection strategies, particularly for large-scale MILP instances.

## 7 CONCLUSIONS

We addressed two pivotal problems in node selection including the sufficient tree representation and assessment of the node quality with delayed rewards. We introduce an innovative tripartite graph representation for the branch-and-bound search tree and provide theoretical evidence. Subsequently, we introduce more comprehensive metrics for node selection and develop a novel TRGNN model, leveraging reinforcement learning to acquire node selection policies. Experiments show that TRGNN significantly outperforms human-designed and learning-based baselines in terms of solving efficiency on three synthetic MILP problems and three real-world MILP problems.

ACKNOWLEDGMENTS

The research is partially supported by National Key R&D Program of China under Grant No. 2021ZD0110400 , Innovation Program for Quantum Science and Technology 2021ZD0302900 and China National Natural Science Foundation with No. 62132018, 62231015, "Pioneer" and "Leading Goose" R&D Program of Zhejiang, 2023C01029, and 2023C01143.

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

# A   THEORETICAL ANALYSIS

We enumerate our theoretical findings and provide detailed proofs to substantiate them.

First, we list the definition of a Markov chain and the data processing inequality (Reza, 1994) to aid in the proof of the subsequent theorem.

**Definition A.1.** *Consider three random variables $X$, $Y$, and $Z$. They form a **Markov chain** $X \to Y \to Z$ if it holds that:*

$$p(x, y, z) = p(x)p(y|x)p(z|y),$$

*where $p(x, y, z)$ is the joint probability density function of $X$, $Y$, and $Z$.*

**Theorem A.1** (Data processing inequality)**.** *If $X \to Y \to Z$ forms a Markov chain, then:*

$$I(X; Y) \geq I(X; Z).$$

**Theorem A.2** (Restatement of Theorem 4.2)**.** *Given a node $N_p$ and its two child nodes $N_l$ and $N_r$, it holds that $I(Q; J_{N_p} \oplus J_{N_l} \oplus J_{N_r}) = I(Q; J_{N_l} \oplus J_{N_r})$, where $J_N$ denotes the extracted feature of the node $N$.*

*Proof of Theorem 4.2.* First, note that $J_{N_l} \oplus J_{N_r} \subseteq J_{N_p} \oplus J_{N_l} \oplus J_{N_r}$, hence $Q \to J_{N_l} \oplus J_{N_r} \to J_{N_p} \oplus J_{N_l} \oplus J_{N_r}$ is a Markov chain. According to the data processing inequality (Theorem A.1), we have:

$$I(Q; J_{N_p} \oplus J_{N_l} \oplus J_{N_r}) \geq I(Q; J_{N_l} \oplus J_{N_r}).$$

To complete the proof, it is necessary to show that:

$$I(Q; J_{N_p} \oplus J_{N_l} \oplus J_{N_r}) \leq I(Q; J_{N_l} \oplus J_{N_r}).$$

Assuming $J_{N_l} \oplus J_{N_r} \to J_{N_p}$, then $Q \to J_{N_p} \oplus J_{N_l} \oplus J_{N_r} \to J_{N_l} \oplus J_{N_r}$ also forms a Markov chain. Thus, by applying the data processing inequality once again, we establish the desired equality.

To prove the Markov chain, we analyze the relationship between the parent node $N_p$ and the child nodes $N_l$ and $N_r$. The node features encompass both the MILP characteristics, denoted by $J_m$, including the variables, constraints and its objective function, and the global attributes within the tree, denoted by $J_g$. Thus, we have: $J_N = (J_m)_N \oplus (J_g)_N$.

In order to prove $J_{N_l} \oplus J_{N_r} \to J_{N_p}$, we only need to prove $(J_m)_{N_l} \oplus (J_m)_{N_r} \to (J_m)_{N_p}$ and $(J_g)_{N_l} \oplus (J_g)_{N_r} \to (J_g)_{N_p}$.

We consider the MILP characteristics, which is only related to their problem structures. The variables and objective function in the branch-and-bound tree is the same as the root node $(J_m)_{N_0}$. We denote the common variables and objective function feature as $CF$, the variable set as $V = \{x_i \mid i \in \{0, 1, \ldots, l\}\}$, with $|V| = l$, and the constraint set of the node $N$ as $S_N$. Thus, we have $(J_m)_N = S_N \oplus CF$.

Because the constraints of both the child nodes $N_l$ and child nodes $N_r$ is the constraint of the parent node $N_p$ added by the branching constraints. Denote $N_p$'s LP relax solution as $x_{N_p}$, and let $z = \lfloor (x_{N_p})_i \rfloor$, where $(x_{N_p})_i$ denotes the value of variable $x_i$ in the LP relax solution $x_{N_p}$.

The branching constraints are denoted as $x_i \leq z$ for $N_l$ and $x_i \geq z + 1$ for $N_r$, where $x_i \in V$ is the selected branch variable on the parent node $N_p$.

Consequently, the sets of constraints for $N_l$ and $N_r$ can be represented as:

$$S_{N_l} = S_{N_p} \cup \{x_i \leq z\},$$

and

$$S_{N_r} = S_{N_p} \cup \{x_i \geq z + 1\}.$$

which leads to a significant overlap: $S_{N_l} \cap S_{N_r} = S_{N_p}$. This implies that the combined MILP characteristics $J_m$ for $N_l$ and $N_r$ encapsulate the MILP characteristics of $N_p$:

$$
\begin{aligned}
(J_m)_{N_l} \cap (J_m)_{N_r} &= (S_{N_l} \oplus CF) \cap (S_{N_r} \oplus CF) \\
&= (S_{N_l} \cap S_{N_r}) \oplus CF \\
&= (S_{N_p} \oplus CF) \\
&= (J_m)_{N_p}.
\end{aligned}
$$

Thus, we establish that a function relationship $f_m$ exists such that: $(J_m)_{N_p} = f_m((J_m)_{N_l} \oplus (J_m)_{N_r})$.

Considering the mutual information equations:

$$
\begin{aligned}
I(Q; J_{N_p} \oplus (J_{N_l} \oplus J_{N_r})) &= I(Q; ((J_m)_{N_p} \oplus (J_g)_{N_p}) \oplus (J_{N_l} \oplus J_{N_r})) \\
&\leq I(Q; f_m((J_m)_{N_l} \oplus (J_m)_{N_r}) \oplus (J_g)_{N_p} \oplus (J_{N_l} \oplus J_{N_r})) \\
&\leq I(Q; ((J_m)_{N_l} \oplus (J_m)_{N_r}) \oplus (J_g)_{N_p} \oplus (J_{N_l} \oplus J_{N_r})) \\
&\leq I(Q; ((J_m)_{N_l} \oplus (J_m)_{N_r}) \oplus (J_g)_{N_p} \oplus ((J_m)_{N_l} \oplus (J_g)_{N_l}) \\
&\quad \oplus ((J_m)_{N_r} \oplus (J_g)_{N_r})) \\
&= I(Q; (J_g)_{N_p} \oplus (J_{N_l} \oplus J_{N_r})).
\end{aligned}
$$

Next, we turn our attention to the global attributes $J_g$, which contain the solving process characteristics of each node and the positional characteristics of each node to represent the whole tree.

The solving process characteristics originates from the an improved feasible integer solution or the LP relaxation solution, denoted as the update of the local upper and lower bounds. We denote the local upper and lower bounds on the node $N$ as $UB_N$ and $LB_N$. We denote the node $N$'s positional characteristics as $p_N$, which can represent the position of the node $N$ in the tree. Thus, we have $(J_g)_N = (LB_N, UB_N) \oplus p_N$.

Given that the LP relaxation solutions of the child nodes, $N_l$ and $N_r$, are also solutions of the node $N_p$. The local upper bound only updates when finding a tighter LP relaxation solution. Thus, it is established that $UB_{N_l} \leq UB_{N_p}$ and $UB_{N_r} \leq UB_{N_p}$. According to Lemma A.1, it is inferred that the infeasible integer solution of the node $N_p$ is also a solution of either $N_l$ or $N_r$. Hence, we deduce that $LB_{N_l} \leq LB_{N_p}$ and $LB_{N_r} \leq LB_{N_p}$. This establishes that:

$$
(LB_{N_p}, UB_{N_p}) \subseteq (LB_{N_l}, UB_{N_l}) \cup (LB_{N_r}, UB_{N_r}).
$$

Since child nodes are derived from the parent node, positional attributes of $p_{N_p}$ of the parent node can be deduced by the two child nodes' positional features $p_{N_l}$ and $p_{N_r}$. We denote a function $f_p$ such that

$$
p_{N_p} = f_p(p_{N_l}, p_{N_r}),
$$

demonstrating that:

$$
(J_g)_{N_p} \subseteq (LB_{N_l}, UB_{N_l}) \cup (LB_{N_r}, UB_{N_r}) \oplus p_{N_p} = (LB_{N_l}, UB_{N_l}) \cup (LB_{N_r}, UB_{N_r}) \oplus f_p(p_{N_l}, p_{N_r}).
$$

Considering the mutual information equations:

$$
\begin{aligned}
I(Q; J_{N_p} \oplus J_{N_l} \oplus J_{N_r}) &\leq I(Q; (J_g)_{N_p} \oplus (J_{N_l} \oplus J_{N_r})) \\
&= I(Q; ((LB_{N_p}, UB_{N_p}) \oplus p_{N_p}) \oplus (J_{N_l} \oplus J_{N_r})) \\
&\leq I(Q; ((LB_{N_l}, UB_{N_l}) \cup (LB_{N_r}, UB_{N_r}) \oplus f_p(p_{N_l}, p_{N_r})) \\
&\quad \oplus (J_{N_l} \oplus J_{N_r})) \\
&\leq I(Q; ((LB_{N_l}, UB_{N_l}) \oplus (LB_{N_r}, UB_{N_r}) \oplus p_{N_l} \oplus p_{N_r}) \\
&\quad \oplus (J_{N_l} \oplus J_{N_r})) \\
&= I(Q; ((J_g)_{N_l} \oplus (J_g)_{N_r}) \oplus (J_{N_l} \oplus J_{N_r})) \\
&= I(Q; J_{N_l} \oplus J_{N_r}).
\end{aligned}
$$

$\square$

**Lemma A.1.** *Given a node $N_p$ and its two child nodes $N_l$ and $N_r$, any infeasible integer solution of node $N_p$ is also an infeasible solution of either $N_l$ or $N_r$.*

*Proof of Lemma A.1.* Suppose there exists an infeasible integer solution $x$ of node $N_0$ that is not an infeasible solution of either child nodes, $N_1$ or $N_2$.

Let $\mathcal{S}_{N_0}$, $\mathcal{S}_{N_1}$, and $\mathcal{S}_{N_2}$ represent the constraint spaces of nodes $N_0$, $N_1$, and $N_2$, respectively. By assumption, $x \in \mathcal{S}_{N_0}$ but $x \notin \mathcal{S}_{N_1} \cup \mathcal{S}_{N_2}$.

Given that $N_1$ and $N_2$ are child nodes of $N_0$, we can express the constraint spaces of $N_1$ and $N_2$ as

$$\mathcal{S}_{N_1} = \mathcal{S}_{N_0} \cap \{x \in \mathcal{C} | x_i \geq z + 1, z \in \mathbb{Z}\}$$

and

$$\mathcal{S}_{N_2} = \mathcal{S}_{N_0} \cap \{x \in \mathcal{C} | x_i \leq z, z \in \mathbb{Z}\}.$$

Thus, $x$ must belong to the set difference $\mathcal{S}_{N_0} \setminus (\mathcal{S}_{N_1} \cup \mathcal{S}_{N_2})$, which is equivalent to

$$\mathcal{S}_{N_0} \cap \{x \in \mathcal{C} | z < x_i < z + 1, z \in \mathbb{Z}\}.$$

However, since $x$ is an integer solution, it cannot belong to the set $\{x \in \mathcal{C} | z < x_i < z + 1, z \in \mathbb{Z}\}$, leading to a contradiction. Therefore, any infeasible integer solution of node $N_0$ must also be an infeasible solution of either $N_1$ or $N_2$. □

**Theorem A.3** (Restatement of Theorem 4.1). *Given the node $N_i$ with features $J_{N_i}$, the tripartite graph representation $\phi_t(F)$ satisfies: $I(Q; \phi_t(F)) = I(Q; F)$, where $F = \bigoplus_{i=0}^{m+n} J_{N_i}$.*

*Proof of Theorem 4.1.* The leaf nodes are denoted as $N_{j_1}, N_{j_2}, \ldots, N_{j_n}$. The set of the leaf nodes' features is denoted as $L = \bigoplus_{i=1}^{n} J_{N_{j_i}}$.

First, let us prove $I(Q; \phi_t(F)) = I(Q; L)$, where $\phi_t(F) = (J_m)_{N_0} \oplus (\bigoplus_{i=1}^{n} B_{N_{j_i}}) \oplus (\bigoplus_{i=1}^{n} (J_g)_{N_{j_i}})$, where $J_N$ denotes the extracted feature of the node $N$, $B_N$ denotes the set of branching constraints added from the root node to the node $N$, and $(J_g)_N$ denotes the global features in the tree of the node $N$, include both the local upper and lower bounds and its positional characteristics.

Thus, we only need to prove $Q \rightarrow \phi_t(F) \rightarrow L$ is a Markov chain.

We have $J_{N_{j_i}} = (J_m)_{N_{j_i}} \oplus (J_g)_{N_{j_i}}$ and $(J_m)_{N_{j_i}} = S_{N_{j_i}} \oplus CF = (S_{N_0} \cup (\bigoplus_{i=1}^{n} B_{N_{j_i}})) \oplus CF = (J_m)_{N_0} \cup ((\bigoplus_{i=1}^{n} B_{N_{j_i}}) \oplus CF)$. Thus, $L = \bigoplus_{i=1}^{n} J_{N_{j_i}} = \bigoplus_{i=1}^{n} (J_m)_{N_0} \cup ((\bigoplus_{i=1}^{n} B_{N_{j_i}}) \oplus CF)) \oplus (J_g)_{N_{j_i}}$.

From $\phi_t(F)$, we can deduce $(J_m)_{N_0}$, $B_{N_{j_i}}$, and $(J_g)_{N_{j_i}}$, for all $i = 1, 2, \ldots, n$. Due to $(J_m)_{N_0} = S_{N_0} \oplus CF$, where $S_{N_0}$ denotes the set of the root node's constraints, we can known $CF$ from $\phi_t(F)$. Then, we can denote there exists a function $f_1$ that $f_1(\phi_t(F)) = L$, which shows $Q \rightarrow \phi_t(F) \rightarrow L$ is a Markov chain.

At the same, we can also known $(J_g)_{N_{j_i}}$, $(J_m)_{N_0}$, and $B_{N_{j_i}}$, for all $i = 1, 2, \ldots, n$, from the representation $L$. Thus, there exists a function $f_2$ that $f_2(L) = \phi_t(F)$, which shows $Q \rightarrow L \rightarrow \phi_t(F)$ is a Markov chain.

Next, we prove $I(Q; \bigoplus_{i=1}^{n} J_{N_{j_i}}) = I(Q; F)$. Because $I(Q; L) \leq I(Q; F)$ obviously satisfies, we only need to prove $I(Q; L) \geq I(Q; F)$. We prove this step by mathematical induction. When expanding the root node for the first time, the conclusion holds according to Theorem 4.2.

Suppose that the conclusion $I(Q; L) \geq I(Q; F)$ holds after the $k-$th selection of nodes. That is, we have $Q \rightarrow L \rightarrow F$ is a Markov chain. We denote the set of all of the nodes in the tree as $\mathcal{T}_k$. When expanding the leaf nodes for the $k + 1-$th time, we denote the selected leaf node as $N_s$. We denote set of the other candidate nodes as $\mathcal{N}'$. Thus, we have $Q \rightarrow (\bigoplus_{N \in \mathcal{N}'} J_N) \oplus J_{N_s} \rightarrow \bigoplus_{N \in \mathcal{T}_k} J_N$ is a Markov chain. We denote the selected node is expanded into two child nodes. To simplify, we denote them as $N_{s,l}$ and $N_{s,r}$. The tree node set turns to be $T_{k+1} = T_k \cup \{N_{s,l}, N_{s,r}\}$. According to Theorem 4.2, we have $Q \rightarrow J_{N_{s,l}} \oplus J_{N_{s,r}} \rightarrow J_{N_s} \oplus J_{N_{s,l}} \oplus J_{N_{s,r}}$ is a Markov chain. Thus, $Q \rightarrow (\bigoplus_{N \in \mathcal{N}'} J_N) \oplus J_{N_{s,l}} \oplus J_{N_{s,r}} \rightarrow \bigoplus_{N \in \mathcal{T}_k} J_N \oplus J_{N_{s,l}} \oplus J_{N_{s,r}}$ is a Markov chain.

According to the Data processing inequality(Theorem A.1), we can prove $I(Q; L) \geq I(Q; F)$.

□

# B MORE DETAILS OF NODE SELECTION PROCESS

**Node selection process.** Algorithm 1 details a general procedure for node selection within this MDP formulation. It describes the primary operations carried out by the node selector in the B&B process. The specific implementation details may vary depending on the chosen node selection strategy, but the fundamental concept remains consistent. Initially, a node is chosen based on the branching strategy. After selection, the node is evaluated, which involves updating its bounds and assessing whether it should be pruned. If the node is not pruned, the tree is branched to enable further exploration.

---

**Algorithm 1** General Node Selection Procedure

---

**Require:** Node list $L$, parent node bounds, probability $Prob$ of heuristic integer solution calculation
1: **for** each node $P$ in $L$ **do**
2:      Select node $P$ according to the node selection strategy.
3:      Update the lower bound of $P$, $LB(P) = lb_{parent}$, where $lb_{parent}$ is the lower bound of parent node of $P$.
4:      Solve the LP relaxation of node $P$, getting the solution $Sol1$ and objective function value $Obj1$.
5:      Update the upper bound of $P$, $UB(P) = Obj1$. Propagate the updated upper bound upward.
6:      **if** $Obj1 \leq LB(P)$ **then**
7:          Prune node $P$.
8:      **else**
9:          **if** $Sol1$ is an integer solution **then**
10:              Update the lower bound of $P$, $LB(P) = Obj1$. Propagate the updated lower bounds upward.
11:          **end if**
12:          Call the heuristic to calculate an integer solution with probability $Prob$, resulting in $Sol2$ and $Obj2$.
13:          **if** $Obj2 > LB(P)$ is an integer solution **then**
14:              Update the lower bound of $P$, $LB(P) = Obj2$. Propagate the updated lower bounds upward.
15:          **end if**
16:          Branch at node $P$ and add the children nodes to list $L$.
17:      **end if**
18: **end for**

---

## C  MORE DETAILS OF TRIPARTITE GRAPH.

**Graph representation.** Having sufficient information is crucial to infer the optimal node for selection. But what qualifies as "sufficient" in this context? An ideal problem representation should be capable of incorporating information that affects node selection, which includes the inherent attributes of the original problem $(A, b, c)$ and the attributes of the explored space. Actually, such problem has a strong mathematical structure (Chen et al., 2022). For instance, if we swap the positions of the $i, j$-th variable in 1, elements in vectors $b, c$ and columns of matrix $A$ will be re-ordered. The reordered features $(\hat{A}, \hat{b}, \hat{c})$ actually represent an exactly equivalent MILP problem with the original one $(A, b, c)$. Such property is named as permutation invariance. If we do not explicitly restrict ML models with a permutation invariant structure, the models may overfit to the variable/constraint orders of instances in the training set. Motivated by this point, we adopt the graph representation that are permutation invariant naturally in Section 4. A list of the features included in our tripartite graph representation is given as Table 5.

Table 5: Description of the constraint, variable and leaf node vertex features, and edge features in our tripartite state representation.

| Category | Feature | Description |
|---|---|---|
| variable vertex | lb | Lower bound. |
| | ub | Upper bound. |
| | objective_coeff | Objective coefficient. |
| | var_type | Type (binary, integer and continuous) as a one-hot encoding. |
| constraint edge | coef | Constraint coefficient. |
| constraint vertex | rhs | Right-hand side of the constraint. |
| | cons_type | Constraint type feature (eq, geq) as a one-hot encoding. |
| leaf node vertex | leaf_lb | Lower bound of the leaf node. |
| | leaf_ub | Upper bound of the leaf node. |
| | depth | Depth of the leaf node. |
| | estimate | Estimate value of the leaf node. |
| branching constraint edge | bc_lb | Lower bound of the variable. |
| | bc_ub | Upper bound of the variable. |

# D  MORE DETAILS OF EXPERIMENTS

The details of experiments are provided in this section.

## D.1  BENCHMARK

**Synthetic datasets.** We similarly employ three synthetic instance families, just like Labassi et al. (2022) in the latest node selection work. The first benchmark is composed of Fixed Charge Multi-commodity Network Flow (FCMCNF) instances (Hewitt et al., 2010), generated from the code of Chmiela et al. (2021). We train and test on instances with $n = 20$ and $m = 1.5 \times n$ commodities, and also evaluate on larger transfer instances with $n = 30$ nodes. The second benchmark is composed of MaxSAT (Ansótegui & Gabàs, 2017) instances, generated following the scheme of Béjar et al. (2009). We train and test on instances with a uniformly sampled number of nodes $n \in [80, 100]$ and transfer on instances with $n \in [120, 150]$. Our third benchmark is composed of Generalized Independent Set (GISP) instances (Colombi et al., 2017), generated from the code of Chmiela et al. (2021). We train and test on instances with a uniformly sampled number of nodes $n \in [60, 70]$ and transfer on instances with $n \in [120, 150]$. All these families require an underlying graph: we use in each case Erdős–Rényi random graphs with the prescribed number of nodes, with edge probability $p = 0.33$ for FCMCNF and $p = 0.66$ for MaxSAT and GISP.

Table 6: Average value of the size of variables and constraints for MILP problems (Train and Test).

| Number | GISP | MaxSAT | FCMCNF | MIK | Corlat | Anonymous |
|---|---|---|---|---|---|---|
| **Avg. Binary** | 65.0 | 1611.70 | 125.61 | 79.17 | 100.0 | 14747.35 |
| **Avg. Int** | 0.0 | 0.0 | 0.0 | 420.83 | 0.0 | 1268.35 |
| **Avg. Continuous** | 625.48 | 0.0 | 3185.67 | 18.33 | 366.0 | 21865.6 |
| **Avg. Cons** | 1249.78 | 1542.16 | 633.36 | 439.17 | 486.46 | 49603.6 |

Table 7: Average value of the size of variables and constraints for MILP problems (Transfer).

| Number | GISP | MaxSAT | FCMCNF |
|---|---|---|---|
| **Avg. Binary** | 134.14 | 3353.61 | 287.23 |
| **Avg. Int** | 0.0 | 0.0 | 0.0 |
| **Avg. Continuous** | 2690.61 | 0.0 | 10754.84 |
| **Avg. Cons** | 5378.52 | 3218.40 | 1410.72 |

**Real-world datasets.** Following He et al. (2014); Nair et al. (2020), our dataset consists of the following components: MIK (Atamtürk, 2003), a set of MILP problems with knapsack constraints, and CORLAT (Gomes et al., 2008), a real dataset used for the construction of a wildlife corridor for grizzly bears in the Northern Rockies region. In addition to these datasets, we have introduced a more challenging dataset into our experiments: the Anonymous problem, sourced from the NeurIPS 2021 ML4CO competition (Gasse et al., 2022). Each problem set is split into training and test sets with 80% and 20% of the instances.

## D.2  BASELINES

The SVM and RankNet methods utilize a multilayer perceptron; the latter varies for one benchmark where they use three hidden layers, while for simplicity, we use a multilayer perceptron with a hidden layer of 32 neurons for all benchmarks (MLP). The GNN method uniquely leverages the structure of the graph to guide node selection. The features used in these papers are roughly similar; again, for simplicity, we adopt the fixed-dimensional features of He et al. (2014) for both the SVM and RankNet method.

## D.3  HYPER-PARAMETER SETTINGS

The hyperparameters primarily consist of those associated with the reward function and the prese-lection size $n$ determined by the "BestEstimate" heuristic algorithm. In our evaluation, we set $n = 5$

and more detailed analysis of how variations in $n$ affect the final node selection results is provided in Section D.7.

**Reward function hyper-parameters.** We denote a parameter $m$ to normalize the path switching steps. The path switching steps are denoted as $s$, the normalized reward for path switching steps is $R_s = \frac{s}{m} - 1$. In this function, if the path switching steps are less than $m$, $R_s > 0$; otherwise, the model receives a penalty. The value of $m$ depends on the size of the problem. We use $m = 15$ for the FCMCNF, MaxSAT, GISP, MIK and CORLAT datasets and $m = 50$ for the Anonymous dataset. Moreover, we take the weight values $w_1 = w_2 = w_3 = \frac{1}{3}$.

Additionally, we follow Labassi et al. (2022) to limit the number of times the learning-based node selector is called. Consistent with the setup in Labassi et al. (2022), the learning-based node selection method is only called before the primal bound is updated four times. After four updates, we switch to the Breadth-First Search method.

## D.4 MORE RESULTS OF NODES

Table 8: Policy evaluation on the synthetic and real-world datasets. The best performance is marked in bold. We report the 1-shifted geometric mean (standard deviation) of the Nodes and Im_N (in percentage).

| Methods | FCMCNF | | MaxSAT | | GISP | |
|---|---|---|---|---|---|---|
| | **Nodes** | **Im_N** | **Nodes** | **Im_N** | **Nodes** | **Im_N** |
| SCIP | 105.58 ± 5.50 | 0.00% | 128.43 ± 2.33 | 0.00% | **100.77 ± 3.32** | **0.00%** |
| SVM | 101.28 ± 4.62 | +3.01% | 224.28 ± 2.01 | -74.63% | 127.15 ± 3.09 | -26.17% |
| RankNet | 92.48 ± 3.70 | +12.40% | 204.54 ± 2.18 | -59.26% | 119.78 ± 3.25 | -18.86% |
| GNN | **78.65 ± 4.21** | **+25.50%** | 194.22 ± 2.31 | -51.22% | 117.56 ± 3.05 | -16.66% |
| TRGNN (Ours) | 230.74 ± 4.50 | -118.54% | **78.31 ± 3.48** | **+39.02%** | 101.07 ± 3.19 | -0.30% |

| Methods | MIK | | CORLAT | | Anonymous | |
|---|---|---|---|---|---|---|
| | **Nodes** | **Im_N** | **Nodes** | **Im_N** | **Nodes** | **Im_N** |
| SCIP | 896.19 ± 4.56 | 0.00% | 273.81 ± 35.66 | 0.00% | 51037.30 ± 31.49 | 0.00% |
| SVM | 870.12 ± 4.64 | +2.90% | 160.23 ± 26.51 | +41.48% | 52369.21 ± 297.75 | -2.60% |
| RankNet | 937.15 ± 5.53 | -4.57% | **156.60 ± 21.65** | **+42.80%** | 53881.66 ± 32.55 | -5.57% |
| GNN | **757.99 ± 4.32** | **+15.42%** | 173.82 ± 22.73 | +36.51% | **16039.64 ± 39.03** | **+68.57%** |
| TRGNN (Ours) | 953.56 ± 4.58 | -6.38% | 276.36 ± 28.88 | -0.93% | 19445.26 ± 36.45 | +61.89% |

Table 9: The generalization ability of TRGNN. The best performance is marked in bold. We report the 1-shifted geometric mean (standard deviation) of the Time.

| Methods | FCMCNF | | MaxSAT | | GISP | |
|---|---|---|---|---|---|---|
| | **Nodes** | **Im_N** | **Nodes** | **Im_N** | **Nodes** | **Im_N** |
| SCIP | 2427.33 ± 6.17 | 0.00% | **390.92 ± 2.66** | **0.00%** | 31552.14 ± 1.89 | 0.00% |
| SVM | 3036.70 ± 5.66 | -25.10% | 582.88 ± 2.88 | -49.10% | **23675.73 ± 2.26** | **+24.96%** |
| RankNet | 1766.20 ± 5.66 | +27.23% | 591.18 ± 2.50 | -51.23% | 31214.54 ± 1.99 | +1.07% |
| GNN | **1710.77 ± 5.21** | **+29.52%** | 503.43 ± 2.07 | -28.78% | 27909.15 ± 2.01 | +11.55% |
| TRGNN (Ours) | 1812.40 ± 8.33 | +25.33% | 531.73 ± 2.72 | -36.02% | 26386.49 ± 2.21 | +16.37% |

Table 8 and Table 9 illustrate the results of various methods solving MILPs, with the number of nodes (Nodes) used as the evaluation metric. To evaluate different node selection methods compared to the default SCIP, we propose an Improvement metric. Specifically, we define the metric by $Im_N = \frac{Nodes(SCIP) - Nodes(M)}{Nodes(SCIP)}$, where $Nodes(SCIP)$ represents the nodes of SCIP, and $Nodes(M)$ represents the nodes of the compared method.

Due to the imposition of a 3600-second time limit, instances exceeding this duration were capped at 3600 seconds for statistical purposes. We provide the results which averaging the node counts at the point of interruption for instances. Some instances in Anonymous dataset could not be solved within

this time frame, rendering comparisons of node counts less meaningful, as RankNet only completed 9/18 and SVM completed 8/18 instances within the time constraint.

Although TRGNN does not consistently yield the smallest tree size, it illustrates a pivotal observation: a reduction in solving time does not invariably equate to a decrease in the number of nodes in the search tree. This reinforces our proposition that the influence of node selection on solving efficiency is multifaceted, intertwining with elements like the cost of path switching, discussed in detail in Section 5.2.

### D.5 RESULTS WITH DIFFERENT SAMPLE SIZES

We conducted tests to assess the impact of training with an increased number of samples. The results demonstrate a consistent upward trend in performance.

Table 10: Comparison of Solving Time (s) for GNN and TRGNN with sample sizes.

| Method | GISP | | MaxSAT | | FCMCNF | |
|---|---|---|---|---|---|---|
| | Test | Transfer | Test | Transfer | Test | Transfer |
| SCIP | 4.27 ± 1.72 | 77.82 ± 1.31 | 11.11 ± 2.22 | 322.73 ± 1.76 | 15.89 ± 1.68 | 2102.49 ± 1.99 |
| GNN_light (250 instances) | 4.57 ± 1.76 | 82.34 ± 1.25 | 11.28 ± 1.36 | 420.44 ± 1.76 | 14.59 ± 1.59 | 2311.61 ± 1.83 |
| GNN (1000 instances) | 4.18 ± 1.44 | 71.53 ± 1.58 | 10.08 ± 1.41 | 400.96 ± 1.86 | 13.57 ± 1.94 | 1968.54 ± 2.19 |
| TRGNN (250 instances) | 2.89 ± 1.40 | 62.02 ± 1.46 | 8.63 ± 1.47 | 300.00 ± 1.72 | **11.62 ± 1.63** | 1410.13 ± 1.74 |
| TRGNN (500 instances) | 2.93 ± 1.40 | **61.98 ± 1.46** | 8.54 ± 1.44 | **299.19 ± 1.71** | 11.65 ± 1.76 | 1417.93 ± 1.74 |
| TRGNN (1000 instances) | **2.81 ± 1.42** | 62.01 ± 1.46 | **8.51 ± 1.40** | 306.80 ± 1.71 | 11.63 ± 1.62 | **1406.12 ± 1.74** |

### D.6 ABLATION STUDY ON TRIPARTITE AND BIPARTITE GRAPH REPRESENTATIONS

With the inclusion of leaf nodes, the tripartite graph has a complexity of $O(|V| + |C| + |LN| + |E| + |V||LN|)$, compared to the bipartite graph's $O(|V| + |C| + |E|)$. While this increases the space overhead, the tripartite representation is more space-efficient during node selection. For every candidate leaf node, the bipartite approach requires a complete graph representation, leading to a space complexity of $O(|LN|(|V| + |C| + |E|))$. In problems where the number of constraints is similar to or exceeds the number of variables, such as GISP and MaxSAT, the tripartite graph saves space by avoiding redundant representations of constraints and variables.

We conducted an ablation experiment on the bipartite graph approach. In this study, we used the BestEstimate heuristic to pre-filter candidate nodes and then represented each candidate node with a bipartite graph within our reinforcement learning framework (BRGNN). The training times for the MaxSAT and GISP datasets are listed in Table 11. The results indicate that the computational overhead primarily arises from reinforcement learning rather than the tripartite graph itself. Moreover, the tripartite graph representation can save 29.37% and 20.16% of training time on MaxSAT and GISP, respectively.

Table 11: Comparison of training time (s) for GNN, BRGNN and TRGNN.

| Method | MaxSAT | GISP |
|---|---|---|
| GNN (IL) | 62.64 | 54.07 |
| BRGNN | 14391.13 | 3658.54 |
| TRGNN (Ours) | 10164.75 | 2921.07 |

We added tests for the average graph construction time, and inference time per instance on the MaxSAT and GISP datasets. Although we acknowledged that reinforcement learning required more time during the training phase compared to imitation learning (IL), it did not add to inference time. Additionally, the tripartite graph approach did not increase additional time overhead compared to the bipartite method.

To demonstrate the impact of the tripartite graph representation on solving efficiency compared to bipartite graphs, we conducted tests on the GISP and MaxSAT datasets. Table 16 shows that

Table 12: Comparison of graph construction time (s) for GNN, BRGNN and TRGNN.

| Method | MaxSAT | GISP |
|---|---|---|
| GNN (IL) | 0.02 | 0.01 |
| BRGNN | 0.01 | 0.00 |
| TRGNN (Ours) | **0.01** | **0.00** |

Table 13: Comparison of inference time(s) for GNN, BRGNN and TRGNN.

| Method | MaxSAT | GISP |
|---|---|---|
| GNN (IL) | 0.16 | 0.05 |
| BRGNN | **0.13** | 0.01 |
| TRGNN (Ours) | 0.20 | **0.01** |

the "BestEstimate" method performs similarly to the default SCIP settings. Our tripartite graph approach (TRGNN) improves over SCIP and BestEstimate. Notably, the tripartite graph method outperforms the bipartite method, achieving improvements of 19.4% on the MaxSAT dataset and 2.5% on the GISP dataset.

Table 14: Comparison of TRGNN and BRGNN on GISP and MaxSAT datasets.

| Method | MaxSAT ($n \in [80, 100]$) | | | GISP ($n \in [60, 70]$) | | |
|---|---|---|---|---|---|---|
| | Time (s) | Wins | Nodes | Time (s) | Wins | Nodes |
| SCIP | 8.15 ± 1.64 | 3/50 | 147.51 ± 2.02 | 3.35 ± 1.30 | 6/50 | 94.81 ± 3.67 |
| BestEstimate | 7.98 ± 1.68 | 11/50 | 177.64 ± 2.13 | 3.53 ± 1.30 | 8/50 | 102.99 ± 3.18 |
| BRGNN | 7.92 ± 1.67 | 5/50 | 200.68 ± 2.05 | 3.20 ± 1.32 | 17/50 | 97.27 ± 3.59 |
| TRGNN (Ours) | **6.38 ± 1.85** | **31/50** | **134.93 ± 2.83** | **3.03 ± 1.27** | **19/50** | **83.72 ± 3.51** |

## D.7 ANALYSIS OF PRE-SELECTION SIZE USING THE HEURISTIC ALGORITHM

In our work, we initially use the heuristic algorithm "BestEstimate" to pre-select a fixed candidate set of size $n$. To determine the potential impact of the heuristic algorithm on solution quality, we increased $n$ from 5 to 10. We recorded the position of the nodes selected by our TRGNN within the rankings provided by the heuristic.

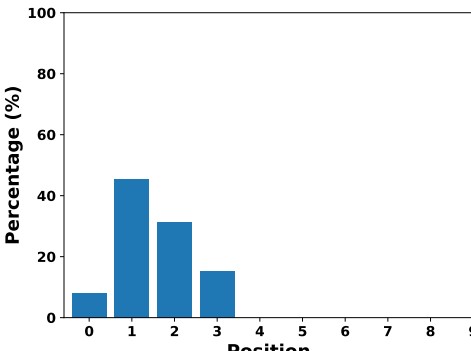
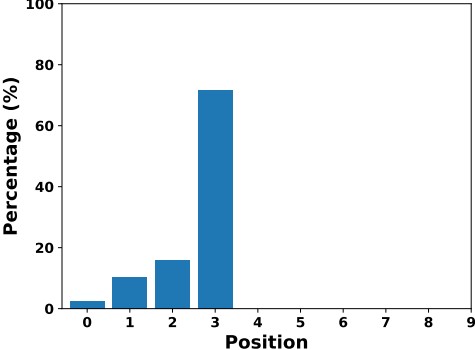

Figure 4: Histograms showing the distribution of positions picked by TRGNN in "BestEstimate," with GISP on the left and MaxSAT on the right.

The results show that in MaxSAT and GISP, nodes ranked higher than 5 by "BestEstimate" are never chosen by TRGNN. This indicates that nodes not ranked among the top by "BestEstimate" have a

very low probability of being selected. This is because "BestEstimate" primarily relies on the node estimate value which represents an optimistic prediction of the best feasible solution that can be found within the subtree of that node. Nodes ranked toward the end generally have poorer estimates, indicating a smaller probability of leading to optimal solutions.

### D.8 Analysis of the impact of the Heuristic Algorithm on the solving results

To ensure fair comparisons, we tested a GNN algorithm based on imitation learning with heuristic pre-selection (GNN+BestEstimate) to eliminate the impact of "BestEstimate" on the solving results. We found that the filtered results were consistent with the original ones because nodes ranked lower by the heuristic "BestEstimate" were seldom chosen by the GNN (see Appendix D.7).

Table 15: Comparison of TRGNN and GNN+BestEstimate on GISP and MaxSAT datasets.

| Method | MaxSAT ($n \in [80, 100]$) | | GISP ($n \in [60, 70]$) | |
|---|---|---|---|---|
| | Time (s) | Nodes | Time (s) | Nodes |
| SCIP | 8.15 ± 1.64 | 147.51 ± 2.02 | 3.12 ± 1.30 | 77.10 ± 3.79 |
| GNN+BestEstimate | 7.71 ± 1.93 | 230.72 ± 3.30 | 3.26 ± 1.35 | 96.99 ± 3.66 |
| TRGNN (Ours) | **6.38 ± 1.85** | **134.93 ± 2.83** | **2.88 ± 1.28** | **80.50 ± 3.39** |

### D.9 Results with the pure Branch-and-Bound framework

While our TRGNN demonstrates significant improvements over the modern solver SCIP, our tripartite graph representation is specifically tailored for the branch-and-bound framework. Our graph does not account for external solver features, such as restarts or cut pools. An intriguing area for future research lies in designing enhanced representations for cutting planes within branch-and-cut algorithms. We believe this is a promising direction.

To better evaluate our node selection method in a branch-and-bound context, we tested SCIP with cutting planes and restarts disabled to better evaluate our node selection method within branch-and-bound. The results showed further improvements for both the GNN and our TRGNN methods compared to SCIP. For example, on MaxSAT, the improvement in solving time for TRGNN over SCIP increased from 7.54% to 12.71%, and on GISP from 13.92% to 18.67%.

Table 16: Comparison of TRGNN and GNN in a branch-and-bound context.

| Method | MaxSAT ($n \in [80, 100]$) | | GISP ($n \in [60, 70]$) | |
|---|---|---|---|---|
| | Time (s) | Nodes | Time (s) | Nodes |
| SCIP | 4.56 ± 1.65 | **78.48 ± 2.66** | 3.16 ± 1.30 | **1119.03 ± 3.90** |
| GNN | 4.42 ± 1.70 | 205.15 ± 3.01 | 2.72 ± 1.18 | 1753.04 ± 1.52 |
| TRGNN (Ours) | **3.98 ± 1.70** | 156.42 ± 3.24 | **2.57 ± 1.17** | 1334.94 ± 1.62 |

