# OpenReview forum: "Learning to Select Nodes in Branch and Bound with Sufficient Tree Representation"
_ICLR.cc/2025/Conference — ICLR 2025 Poster_

### Official Review · Reviewer_zdY7 · 2024-10-16

**Soundness:** 3
**Presentation:** 3
**Contribution:** 3
**Rating:** 6
**Confidence:** 4

**Summary:**

The paper presents a novel approach to node selection in Branch-and-Bound (B&B) algorithms for Mixed Integer Linear Programming (MILP). It proposes using a Tripartite Graph Neural Network (TRGNN) combined with reinforcement learning to address two major challenges: sufficient representation of the B&B tree and comprehensive node quality assessment. The tripartite graph representation effectively captures tree information, while the reinforcement learning framework uses delayed rewards to optimize node selection. Experiments show that TRGNN significantly outperforms traditional human-designed and learning-based approaches in solving efficiency, achieving notable improvements across a variety of MILP benchmarks.

**Strengths:**

### 1. **Innovative Representation Approach**:
   - The use of a tripartite graph representation to capture information about the entire branch-and-bound (B&B) tree is an original contribution. The approach is theoretically proven to be sufficient, ensuring that the representation fully captures the relevant information needed for effective node quality assessment.

### 2. **Reinforcement Learning for Node Selection**:
   - The paper employs reinforcement learning (RL) to learn an effective node selection policy, which is a significant improvement over traditional imitation learning methods. By incorporating delayed rewards, the RL-based approach captures the long-term impact of node selection decisions, leading to better outcomes.

### 3. **Comprehensive Evaluation Metrics**:
   - The introduction of three metrics—global gap updates, potential for optimality, and path-switching overhead—provides a more comprehensive assessment of node quality. This helps the TRGNN model make more informed decisions during the B&B process.

### 4. **Strong Experimental Results**:
   - Experiments demonstrate that TRGNN significantly outperforms both human-designed heuristics and other learning-based methods, including GNN-based models and traditional SVM approaches. TRGNN achieves up to 59.42% reduction in solving time, indicating its practical impact on MILP problem-solving efficiency.

### 5. **Generalization Capability**:
   - TRGNN's ability to generalize to larger MILP instances beyond those used in training is a key strength. The model outperforms existing solvers on significantly larger problem sizes, which underscores the scalability and robustness of the proposed approach.

### 6. **Theoretical Grounding**:
   - The paper provides solid theoretical foundations for its approach, including proofs that the tripartite graph representation is sufficient in information theory. This enhances the credibility of the methodology and helps justify its effectiveness in solving the MILP node selection problem.

### 7. **Efficient Learning**:
   - The TRGNN model demonstrates higher learning efficiency compared to conventional models, requiring fewer training samples to achieve comparable performance. This efficiency is beneficial in scenarios where data collection is costly or time-consuming.

**Weaknesses:**

### 1. **Complexity of Graph Representation**:
   - **Weakness**: The tripartite graph representation is complex, especially for large-scale instances, which may lead to high computational costs.
   - **Suggestion**: Consider providing an analysis of the computational overhead introduced by the tripartite graph representation. Discuss possible optimizations or alternative representations that balance complexity and computational efficiency.

### 2. **Limited Baseline Comparisons**:
   - **Weakness**: The experimental comparisons focus primarily on human-designed heuristics and learning-based models but do not include more recent or diverse approaches in GNN-based optimization or other advanced reinforcement learning methods.
   - **Suggestion**: Expand the list of baseline models to include state-of-the-art approaches in GNNs and other deep learning-based optimization techniques. This would provide a more comprehensive evaluation of TRGNN's advantages.

### 3. **Scalability Concerns**:
   - **Weakness**: Although the model shows good performance on larger problem instances, there is insufficient discussion on the scalability of the method concerning memory requirements and computational resources.
   - **Suggestion**: Include an analysis of memory consumption and computational requirements when scaling the model to large MILP instances. This would help understand the practical limitations and requirements for deploying the approach in real-world scenarios.

### 4. **Lack of Ablation Study on Graph Representation**:
   - **Weakness**: The paper does not thoroughly evaluate the effectiveness of the tripartite graph representation versus other potential graph representations (e.g., bipartite or simpler versions).
   - **Suggestion**: Conduct an ablation study comparing the tripartite graph representation with other simpler graph structures to demonstrate why it is more effective in capturing relevant information for node selection.

### 5. **Limited Analysis of Hyperparameters**:
   - **Weakness**: There is little to no discussion on the sensitivity of the model's hyperparameters, such as the reinforcement learning reward weights and the parameters in the graph convolution layers.
   - **Suggestion**: Provide a sensitivity analysis of key hyperparameters, explaining how they impact the model's performance. This would help practitioners effectively tune the model for different MILP problems.

### 6. **Implementation and Practical Considerations**:
   - **Weakness**: The paper lacks detailed information on implementation aspects, such as training time, hardware requirements, and challenges in real-world implementation.
   - **Suggestion**: Add practical details about the model's implementation, including the training time required for various datasets, hardware used, and potential challenges encountered during implementation. This would aid in understanding the practical feasibility of adopting TRGNN.

### 7. **Limited Applicability to Non-MILP Problems**:
   - **Weakness**: The paper focuses exclusively on MILP problems and does not discuss whether the proposed approach could be generalized to other types of optimization problems.
   - **Suggestion**: Include a discussion on the potential for adapting TRGNN to other combinatorial optimization problems, such as Mixed-Integer Nonlinear Programming (MINLP) or general constraint satisfaction problems. This would extend the relevance of the proposed method.

**Questions:**

See above.

---

> ### Author Response · Authors · 2024-11-20
> **Response to Reviewer zdY7 (Part 1/2)**
>
> Thanks for your valuable feedback and suggestions. We will address the issues pointed out in the weakness section in order.
> ### **Response to the weaknesses 1:**
> With the inclusion of leaf nodes, the tripartite graph has a complexity of O(|V|+|C|+|LN|+|E|+|V||LN|), compared to the bipartite graph's O(|V|+|C|+|E|). While this increases the space overhead, the tripartite representation is more space-efficient during node selection. For every candidate leaf node, the bipartite approach requires a complete graph representation, leading to a space complexity of O(∣LN∣(|V|+|C|+|E|)). In problems where the number of constraints is similar to or exceeds the number of variables, such as GISP and MaxSAT, the tripartite graph saves space by avoiding redundant representations of constraints and variables.
>
> We conducted an ablation experiment on the bipartite graph approach and included the results in Appendix E.6 of the revised manuscript. In this study, we used the BestEstimate heuristic to pre-filter candidate nodes and then represented each candidate node with a bipartite graph within our reinforcement learning framework (BRGNN). The training times for the MaxSAT and GISP datasets are listed below. The results indicate that the computational overhead primarily arises from reinforcement learning rather than the tripartite graph itself. Moreover, the tripartite graph representation can save 29.37% and 20.16% of training time on MaxSAT and GISP, respectively.
>
> Training time(s)
> | Method | MaxSAT  |GISP |
> |-------|-----|------|
> | GNN(IL) | 62.64  |  54.07 |
> | BRGNN | 14391.13  |3658.54  |
> | TRGNN(Ours) |  10164.75  | 2921.07 |
>
> ### **Response to the weaknesses 2:**
> We compared state-of-the-art GNN-based approaches for node selection. Results can be found in Tables 2 and 4. Compared to GNN policy, TRGNN demonstrates notable efficiency improvements across all tested problems, being 26.13% faster in FCMCNF, 5.71% in MaxSAT, 20.29% in GISP, 70.18% in MIK, 10.19% in CORLAT, and 61.06% in Anonymous.
> ### **Response to the weaknesses 3:**
> To address your concern, we have measured and reported the computational time for both graph construction and model inference separately, as these components account for the majority of the additional computational requirements introduced by our method. These times are included in the overall solving time, but we have listed them separately here to highlight the computational overhead of the TRGNN model:
>
> Graph Construction Time(s)
> | Method | MaxSAT(Transfer) |GISP(Transfer) |
> |-------|-----|------|
> | GNN | 0.08  |  0.09 |
> | TRGNN(Ours) |  0.04  | 0.02 |
>
> Inference Time(s)
> | Method | MaxSAT (Transfer) |GISP (Transfer)|
> |-------|-----|------|
> | GNN | 1.61  |  0.43 |
> | TRGNN(Ours) |  1.22  | 0.25 |
>
> These results show that both the graph construction time and inference time are relatively small compared to the overall solving time, even for larger problem instances.
> ### **Response to the weaknesses 4:**
> To demonstrate the impact of the tripartite graph representation on solving efficiency compared to bipartite graphs, we conducted an ablation study. In this study, we used the BestEstimate heuristic to pre-filter candidate nodes and then represented each candidate node with a bipartite graph within our reinforcement learning framework (BRGNN).
>
> We conducted tests on the GISP and MaxSAT datasets. Experiment shows that the BestEstimate method performs similarly to the default SCIP settings. Our tripartite graph approach (TRGNN) improves over SCIP and BestEstimate. Notably, the tripartite graph method outperforms the bipartite method, achieving improvements of 19.4% on the MaxSAT dataset and 2.5% on the GISP dataset.
>
> The results are as follows:
>
> MaxSAT($n\in[80,100]$)
> | Method | time(s)  |wins |nodes |
> |-------|-----|------|----|
> | SCIP      | 8.15  ± 1.64 |3/50 |147.51  ± 2.02|
> | BestEstimate | 7.98  ± 1.68|11/50  |177.64  ± 2.13|
> | BRGNN |  7.92  ± 1.67 |5/50 |200.68  ± 2.05  |
> | TRGNN(Ours) |  **6.38  ± 1.85** |**31/50**  |**134.93  ± 2.83**  |
>
> GISP($n\in[60,70]$)
> | Method | time(s) |wins  |nodes |
> |-------|-----|----|-----|
> | SCIP      |  3.35  ± 1.30| 6/50 |94.81  ± 3.67|
> | BestEstimate |  3.53  ± 1.30 |8/50  |102.99  ± 3.18|
> | BRGNN | 3.20  ± 1.32 | 17/50 |97.27  ± 3.59  |
> | TRGNN(Ours) |  **3.03  ± 1.27** |**19/50**  |**83.72  ± 3.51**  |

---

> > ### Author Response · Authors · 2024-11-20
> > **Response to Reviewer zdY7 (Part 2/2)**
> >
> > ### **Response to the weaknesses 5:**
> > Our GNN training approach aligns with standard practices, and extensive research on GNN hyperparameters already exists. The key hyperparameter in our work is n from the "BestEstimate" heuristic. We conducted additional experiments to better understand this effect. In our work, we initially use the heuristic algorithm "BestEstimate" to pre-select a fixed candidate set of size n. To determine the potential impact of the heuristic algorithm on solution quality, we increased n from 5 to 10. We recorded the position of the nodes selected by our TRGNN within the rankings provided by the heuristic. We included these graphs in Appendix E.7 of the revised manuscript.
> >
> > The results show that in MaxSAT and GISP, nodes ranked higher than 5 by BestEstimate are never chosen by TRGNN. This suggests that TRGNN's effectiveness is not heavily reliant on the candidate set size, as long as n is sufficiently large (e.g., $n=5$). Results indicate that nodes not ranked among the top by "BestEstimate" have a very low probability of being selected. This is because BestEstimate primarily relies on the node estimate value which represents an optimistic prediction of the best feasible solution that can be found within the subtree of that node. Nodes ranked toward the end generally have poorer estimates, indicating a smaller probability of leading to optimal solutions.
> > ### **Response to the weaknesses 6:**
> > The training times for the MaxSAT and GISP datasets are listed below. The results indicate that the computational overhead primarily arises from reinforcement learning rather than the tripartite graph itself. Moreover, the tripartite graph representation can save 29.37% and 20.16% of training time on MaxSAT and GISP, respectively.
> >
> > Training time(s)
> > | Method | MaxSAT  |GISP |
> > |-------|-----|------|
> > | GNN(IL) | 62.64  |  54.07 |
> > | BRGNN | 14391.13  |3658.54  |
> > | TRGNN(Ours) |  10164.75  | 2921.07 |
> >
> > Hardware requirements are detailed on page 9, line 456. The training process was conducted on a single machine equipped with eight NVIDIA GeForce RTX 4090 GPUs and two AMD EPYC 7763 CPUs.
> >
> > ### **Response to the weaknesses 7:**
> > Thank you for your suggestion. Our current approach is tailored for MILP problems using branch-and-bound, which isn’t typically applicable to CSPs, as they don’t primarily use branch-and-bound methods. However, we recognize that some nonlinear problems, like MINLP, do employ branch-and-bound techniques, offering potential for adaptation. However, there are challenges, such as adapting the tripartite graph representation to capture nonlinear constraints effectively. Exploring these extensions is a promising direction for future research, and we plan to investigate this further.

---

> > > ### Comment · Reviewer_zdY7 · 2024-11-26
> > >
> > > Thanks for your great repy and I raise my score.

---

> > > > ### Author Response · Authors · 2024-11-27
> > > >
> > > > Dear Reviewer,
> > > >
> > > > We sincerely appreciate the time and effort you spent on our work. Thank you!
> > > >
> > > > Best regards,
> > > >
> > > > Authors

---

### Official Review · Reviewer_bpuZ · 2024-10-31

**Soundness:** 3
**Presentation:** 3
**Contribution:** 3
**Rating:** 8
**Confidence:** 5

**Summary:**

This work presents a novel way of selecting nodes in the Branch-and-Bound algorithm, by constructing a sufficient representation that splits the features into “global” features containing information of the root MILP, and “leaf” features containing information on each individual leaves. This allows them to represent the entire tree, while keeping the representation compact by bundeling all features of the principal relaxation into a common bipartite graphs and only representing the changes towards one of the leaf nodes.

**Strengths:**

Strengths:
-	Well motivated representation via Mutual information
-	Compact representation
-	High benchmark performance
-	The reward function involving path switching is generally interesting for all node selection methods

I also want to commend the authors for noting the difference in performance of different SCIP versions which is often neglected in comparisons between methods.

**Weaknesses:**

One notable issue in the sufficiency argument of the tripartite representation is the assumption of a Markov mapping between parent and child, which is not generally true: For instance, restarts or cut pools may add arbitrary external information that is not located in the parent into the child. This means that in modern Branch-and-Bound solvers such as SCIP, one can generally not get the decompositions shown in appendix B.
In fact, the method seems to have no way of dealing with cutting planes (it does not even contain features to explain cutting planes).

I recommend qualifying the sufficiency conditions with conditions making clear that these representations are only valid for traditional Branch-and-Bound methods, not more modern restarting Branch-and-Cut methods such as SCIP.

A more general problem is found in the benchmarking: This work trains a node selector on some problem XYZ and then benchmarks on different instances of the same problem XYZ. This is a flawed way of doing benchmarking: If you want to compare a _general_ node selector (SCIP) against your node selector, you also have to use a general problem set.

As is, the quality of the heuristic is not really meaningful: Generally, humans can also develop better heuristics if the type of problem is known. To show that your method leads to superior performance in one of these tasks (e.g. MAXSAT) you would need to compare against an exact MAXSAT solver. Alternatively, you could also benchmark a single selector on all domains, to show that your method can indeed compete against the general node selectors developed in e.g. SCIP.

The statement “Experiments show that TRGNN significantly outperforms human-designed and learning-based baselines” is therefore not supported, since both methods are evaluated under different standards (SCIP “one size fits all” vs. learned ones for every instance type).

Can you re-run the experiments with a single node selector for all tasks?


Figure 2. is hard to read due to the apparent circular dependency of “n nodes”, TRGNN and the “Q-values”. The figure only makes sense if taken in context with the main text: My recommendation: Remove the arrow from the Q-value vector to the “n nodes” bubble, add numbers to the arrows, and explain the figure step-by-step in the description (the figure should be understandable without needing additional context).


For related work, there are two papers that already include a full tree representation “Learning to branch with Tree MDPs “ https://arxiv.org/pdf/2205.11107 uses a tree-like MDP to do branching, and “Reinforcement Learning for Node Selection in Branch-and-Bound” https://arxiv.org/abs/2310.00112 uses a tree-like state representation fornode selection.

Do you have an ablation of the impact of your path-switching reward? While path switching is a measurable effect in node selection. There are many other aspects as well, like the complexity of a node is not being constant: Depending on how many (dual) simplex iterations you need to restore optimality under the new constraints, and which heuristics are run for how long (e.g. diving heuristics) the cost of exploring a node can vary significantly.

What are the effects of the truncation to only the top-n “best estimate” nodes? I understand you can’t test this hyperparameter extensively, but you could give a distribution over which position in the "best node list" is picked by your selector. If you record this on a histogram, you could estimate the impact of truncating the top-n to a k<n.
This would also show how much your method depends on the quality of the "best estimate" node selector.

I generally like this work, with the one big holdup being the lack of generalization of a learned node selector to different problem sets.

**Questions:**

- Can you re-run the experiments with a single node selector for all tasks?
- Regarding experiments in table 2: In line 466 you say you use geometric means and standard deviations, while table 2 claims it uses arithmetic means. Which one is used in the table?
- Do you have an ablation of the impact of your path-switching reward?
- What are the effects of the truncation to the top-n “best estimate” nodes?

See "weaknesses" for more details.

---

> ### Author Response · Authors · 2024-11-20
> **Response to Reviewer bpuZ (Part 1/2)**
>
> Thanks for your valuable feedback and suggestions. We will address the issues pointed out in the weakness section in order.
> ### **Response to the weaknesses 1:**
> Our work is indeed more suited to branch-and-bound algorithms rather than branch-and-cut. Based on your valuable suggestion, we tested SCIP with cutting planes and restarts disabled to better evaluate our node selection method within branch-and-bound.
> The results are included in Appendix E.9 of the revised manuscript.
>
> MaxSAT($n\in[80,100]$)
> | Method | time(s)  |nodes |
> |-------|-----|------|
> | SCIP      | 4.56  ± 1.65 |**78.48  ± 2.66**|
> | GNN |  4.42  ± 1.70 |205.15  ± 3.01  |
> | TRGNN(Ours) |  **3.98  ± 1.70** |156.42  ± 3.24 |
>
> GISP($n\in[60,70]$)
> | Method | time(s)  |nodes |
> |-------|-----|------|
> | SCIP      |  3.16  ± 1.30|**119.03  ± 3.90**|
> | GNN |2.72  ± 1.18 |1753.04  ± 1.52  |
> | TRGNN(Ours)|  **2.57  ± 1.17** |1334.94  ± 1.62 |
>
> The results show further improvements for both the GNN and our TRGNN methods compared to SCIP. For example, on MaxSAT, the improvement in solving time for TRGNN over SCIP increases from 7.54% to 12.71%, and on GISP from 13.92% to 18.67%. Additionally, we find that although both BRGNN and TRGNN resulted in more nodes for the GISP problem, the solving time actually decreased. This further validates our observation that simply finding the optimal solution faster, thus excluding other nodes, is not enough to guide efficient node selection.
> ### **Response to the weaknesses 2 and question 1:**
> We acknowledge that for specialized problems, tailored heuristics can achieve superior performance. However, designing optimal heuristics for each class of optimization problems often requires domain-specific expertise and extensive exploration. Our work focuses on enhancing the node selection module within the general framework of MILP solvers, specifically optimizing its performance through learning-based methods.
>
> We believe that comparing our method against SCIP is reasonable, as our work builds upon SCIP’s benchmark framework and demonstrates the potential of learning-based node selection approaches to surpass SCIP in specific problem domains. Nonetheless, we fully agree with your suggestion to evaluate our method on a broader set of tasks to assess its generalization capability. we constructed a dataset that combines multiple problem types, including GISP and MaxSAT, and re-ran the experiments with a single TRGNN model trained across all tasks. The results are as follows:
>
> Times(s)
> | Method | GISP  | MaxSAT |
> |-------|-------|-------|
> | SCIP |  3.10  ± 1.34|8.88  ± 1.22|
> | TRGNN(Single) |  2.71  ± 1.18 |8.51  ± 1.28  |
> | TRGNN |  **2.67  ± 1.19** |**6.53  ± 1.30**  |
>
> These findings demonstrate that even with a single trained model across multiple problem types, our approach remains competitive against SCIP. However, we found that when training on a general dataset, the performance is not as strong as training specifically for each problem. We believe your suggestion of using a single model to accommodate all problem types is fascinating, and we plan to explore it further. Achieving this goal involves overcoming several challenges, such as designing a model that can accurately learn problem features to provide suitable node selection strategies within similar problem types, ultimately achieving solving times comparable to those trained on specific problem datasets.
>
> We also revised the experimental description to ensure precision. We now specifically state that our method outperforms other node selection strategies within the branch-and-bound framework, rather than making a general claim about heuristic superiority.
>
> ### **Response to the weaknesses 3:**
> We have enhanced the figure with a step-by-step description to ensure it is self-contained and more easily understood. We first apply the heuristic algorithm ``BestEstimate'' to pre-select $n$ candidate nodes and incorporate their estimate values into the node features. Subsequently, the TRGNN model processes these features and outputs a Q-value vector, from which the nodes with the highest Q-values are selected as the final choices.
>
> ### **Response to the weaknesses 4:**
> Thank you for pointing out that our literature review could be improved. We have carefully read these two papers and included descriptions of their methods in the related work section (p3,l122).

---

> > ### Author Response · Authors · 2024-11-20
> > **Response to Reviewer bpuZ (Part 2/2)**
> >
> > ### **Response to the weaknesses 5 and question 3:**
> > We conducted an ablation study to evaluate the impact of different reward components in our framework. As shown in Table 3, we compare the performance of TRGNN, TRGNN-1, and TRGNN-2 under varying reward strategies across four datasets: FCMCNF, GISP, MIK, and Anonymous. Specifically, TRGNN-1 uses only the node’s potential to lead to an optimal solution (R_o), while TRGNN-2 incorporates both the global gap (R_g) and the optimal potential reward (R_o). Our full model, TRGNN, additionally considers the path-switching overhead (R_s). The results clearly demonstrate that TRGNN outperforms both TRGNN-1 and TRGNN-2, indicating that both the gap update and the path-switching overhead are crucial for efficient exploration in complex tasks.
> >
> > Your feedback about the varying costs of exploring nodes is insightful. We agree that exploration costs, such as the number of (dual) simplex iterations required to restore optimality, the heuristics applied (e.g., diving heuristics), and their runtime, are highly variable and can significantly influence solver performance. Compared to previous work, we emphasize one such critical factor: the path-switching overhead. In future work, we aim to investigate other potential contributors to exploration costs to provide a more comprehensive understanding of these dynamics.
> >
> > ### **Response to the weaknesses 6 and question 4:**
> > Thank you for the excellent suggestion regarding analyzing the impact of the hyperparameter parameter $n$ in the “BestEstimate” heuristic. We conducted additional experiments to better understand this effect. To evaluate the potential impact of $n$, we initially conducted experiments with $n=5$ , as described in the main text. To further investigate how varying $n$ influences our results and reduce the potential bias introduced by the candidate set, we expanded $n$ to 10 and analyzed the distribution of the nodes selected by TRGNN within the rankings provided by the BestEstimate heuristic. These results are visualized in histograms included in Appendix E.7.
> >
> > The results show that in MaxSAT and GISP, nodes ranked higher than 5 by BestEstimate are never chosen by TRGNN. This indicates that nodes not ranked among the top by "BestEstimate" have a very low probability of being selected. This is because BestEstimate primarily relies on the node estimate value which represents an optimistic prediction of the best feasible solution that can be found within the subtree of that node. Nodes ranked toward the end generally have poorer estimates, indicating a smaller probability of leading to optimal solutions.
> > ### **Response to the question 2:**
> > This was a typo in the text. The results in Table 2 use 1-shifted geometric means, not arithmetic means.

---

> > > ### Comment · Reviewer_bpuZ · 2024-11-21
> > >
> > > Thank you for your thorough response. All my concerns were addressed sufficiently and I will increase my score to match.

---

> > > > ### Author Response · Authors · 2024-11-27
> > > >
> > > > Thank you for the kind words and your reviewing efforts!

---

### Official Review · Reviewer_c3Ua · 2024-11-02

**Soundness:** 3
**Presentation:** 3
**Contribution:** 3
**Rating:** 6
**Confidence:** 4

**Summary:**

The paper addresses challenges in node selection within the branch-and-bound (B&B) algorithm for Mixed Integer Linear Programming (MILP). It introduces two key contributions to improve node selection effectiveness: (1) a novel tripartite graph representation of B&B trees, and (2) a deep reinforcement learning approach that treats node selection as a learning-based strategy, enabling the model to optimize node selection by training on a set of instances.

**Strengths:**

(1) This paper introduces a novel tripartite graph representation of B&B trees, theoretically proven to contain sufficient data for optimal node selection. This representation integrates the problem structure of MILP, typically represented as a bipartite graph, with the global information of the B&B algorithm.

(2) This paper analyzes the shortcomings of prior works, framing node selection as a learning-based strategy and employing reinforcement learning to identify an optimal node selection heuristic that surpasses expert-designed heuristics.

**Weaknesses:**

(1) It is well-known that B&B trees inherently contain both the detailed information of the B&B algorithm and the problem structure of MILP. This raises the question of whether it is necessary to construct a tripartite graph to represent B&B trees, given that this approach increases the graph scale, resulting in greater storage and computational complexity. The authors should analyze how the advantages of the tripartite representation compensate for these increased demands. Additionally, it would be helpful to assess the specific contributions of the MILP bipartite graph within the tripartite structure and quantify its impact on the representation's effectiveness.

(2) In the MDP process, each action requires a heuristic algorithm to generate candidate nodes. It would be helpful to know the time complexity of this heuristic algorithm and whether its solution quality is affected by the increased scale of MILP instances.

(3) When selecting a specialized solver for MILP, it is advisable to consider higher-performance options like Gurobi, as this can enhance the credibility of the results. Additionally, the baseline methods used are not the most recent; the latest state-of-the-art approaches for solving MILP should be considered, rather than limiting the focus solely to node selection. Other strategic methods should also be included in the evaluation.

**Questions:**

(1) Please analyze the time complexity of the TRGNN and other baseline methods.
(2) The reinforcement learning framework claims improved node selection by minimizing delayed rewards; however, how are potential biases in the reward function, such as those from pre-selecting candidate nodes, accounted for to ensure fair comparisons with existing machine learning baselines?

---

> ### Author Response · Authors · 2024-11-20
> **Response to Reviewer c3Ua (Part 1/2)**
>
> Thanks for your valuable feedback and suggestions. We will address the issues pointed out in the weakness section in order.
> ### **Response to the weaknesses 1 and the question 1:**
> **Complexity**: With the inclusion of leaf nodes, the tripartite graph has a complexity of O(|V|+|C|+|LN|+|E|+|V||LN|), compared to the bipartite graph's O(|V|+|C|+|E|). While this increases the space overhead, the tripartite representation is more space-efficient during node selection. For every candidate leaf node, the bipartite approach requires a complete graph representation, leading to a space complexity of O(|LN|(|V|+|C|+|E|)). In problems where the number of constraints is similar to or exceeds the number of variables, such as GISP and MaxSAT, the tripartite graph saves space by avoiding redundant representations of constraints and variables.
>
> We conducted an ablation experiment on the bipartite graph approach and included the results in Appendix E.6 of the revised manuscript. In this study, we used the BestEstimate heuristic to pre-filter candidate nodes and then represented each candidate node with a bipartite graph within our reinforcement learning framework (BRGNN). The training times for the MaxSAT and GISP datasets are listed below. The results indicate that the computational overhead primarily arises from reinforcement learning rather than the tripartite graph itself. Moreover, the tripartite graph representation can save 29.37% and 20.16% of training time on MaxSAT and GISP, respectively.
>
> Training time(s)
> | Method | MaxSAT  |GISP |
> |-------|-----|------|
> | GNN(IL) | 62.64  |  54.07 |
> | BRGNN | 14391.13  |3658.54  |
> | TRGNN(Ours) |  10164.75  | 2921.07 |
>
> We added tests for the average graph construction time, and inference time per instance on the MaxSAT and GISP datasets. Although we acknowledged that reinforcement learning required more time during the training phase compared to imitation learning (IL), it did not add to inference time. Additionally, the tripartite graph approach did not increase additional time overhead compared to the bipartite method.
>
> Graph Construction Time(s)
> | Method | MaxSAT  |GISP |
> |-------|-----|------|
> | GNN(IL) | 0.02  |  0.01 |
> | BRGNN | 0.01  | 0.00 |
> | TRGNN(Ours) |  0.01  | 0.00 |
>
> Inference Time(s)
> | Method | MaxSAT  |GISP |
> |-------|-----|------|
> | GNN(IL) | 0.16  |  0.05 |
> | BRGNN | 0.13  | 0.01 |
> | TRGNN(Ours) |  0.20  | 0.01 |
>
> **Advantages of the tripartite representation and contributions of the bipartite graph**: The following results demonstrate the advantages of the tripartite graph over the bipartite representation. Experiments show that both TRGNN and BRGNN provide improvements over SCIP. Notably, the tripartite graph method outperforms the bipartite method, achieving improvements of 19.4% on the MaxSAT dataset and 2.5% on the GISP dataset.
>
> MaxSAT($n\in[80,100]$)
> | Method | time(s)  |wins |nodes |
> |-------|-----|------|----|
> | SCIP      | 8.15  ± 1.64 |3/50 |147.51  ± 2.02|
> | BestEstimate | 7.98  ± 1.68|11/50  |177.64  ± 2.13|
> | BRGNN |  7.92  ± 1.67 |5/50 |200.68  ± 2.05  |
> | TRGNN(Ours) |  **6.38  ± 1.85** |**31/50**  |**134.93  ± 2.83** |
>
> GISP($n\in[60,70]$)
> | Method | time(s) |wins  |nodes |
> |-------|-----|----|-----|
> | SCIP      |  3.35  ± 1.30| 6/50 |94.81  ± 3.67|
> | BestEstimate |  3.53  ± 1.30 |8/50  |102.99  ± 3.18|
> | BRGNN | 3.20  ± 1.32 | 17/50 |97.27  ± 3.59  |
> | TRGNN(Ours) |  **3.03  ± 1.27** |**19/50**  |**83.72  ± 3.51**  |
> ### **Response to the weaknesses 2:**
> **Time complexity of this heuristic algorithm**: The heuristic algorithm "BestEstimate" operates efficiently, requiring less than $1×10^{−5}$ seconds per instance on GISP and less than $1×10^{−3}$ seconds on MaxSAT.
>
> **Impact of the heuristic algorithm**:  In our work, we initially use the heuristic algorithm "BestEstimate" to pre-select a fixed candidate set of size n. To determine the potential impact of the heuristic algorithm on solution quality, we increased n from 5 to 10. We recorded the position of the nodes selected by our TRGNN within the rankings provided by the heuristic. These results are visualized in histograms included in Appendix E.7.
>
>  The results show that in MaxSAT and GISP, nodes ranked higher than 5 by BestEstimate are never chosen by TRGNN. This indicates that nodes not ranked among the top by "BestEstimate" have a very low probability of being selected. This is because BestEstimate primarily relies on the node estimate value which represents an optimistic prediction of the best feasible solution that can be found within the subtree of that node. Nodes ranked toward the end generally have poorer estimates, indicating a smaller probability of leading to optimal solutions.

---

> ### Author Response · Authors · 2024-11-20
> **Response to Reviewer c3Ua (Part 2/2)**
>
> ### **Response to the weaknesses 3:**
> We did not perform comparisons using Gurobi because it is a commercial solver that does not permit external modifications to the node selection module. We chose SCIP because it is an established benchmark solver widely adopted in the field for evaluating the performance of different components within branch-and-bound algorithms, as demonstrated in studies on node [1], variable [2,3], and cut-plane selection [4].
>
> This work specifically focuses on the node selection problem. Although methods based on reinforcement learning have been applied to tasks such as variable selection and cut-plane selection within branch-and-bound algorithms, these components are distinct modules and do not directly relate to one another. The main challenges faced by these problems are unique, and each can be studied independently. For example, variable selection involves only representing the characteristics of a single node to identify the best branching variables, without the need to consider multiple nodes or global tree attributes. Additionally, there are theoretically proven optimal branching strategies, such as strong branching, which can be effectively learned. However, the criteria for node selection remain ambiguous. We believe that making comparisons between these disparate aspects is unnecessary.
>
> [1] Labassi A G, Chételat D, Lodi A. Learning to compare nodes in branch and bound with graph neural networks[J]. Advances in neural information processing systems, 2022, 35: 32000-32010.
>
> [2] Lin J, Meng X U, Xiong Z, et al. CAMBranch: Contrastive Learning with Augmented MILPs for Branching[C]//The Twelfth International Conference on Learning Representations.
>
> [3] Gasse M, Chételat D, Ferroni N, et al. Exact combinatorial optimization with graph convolutional neural networks[J]. Advances in neural information processing systems, 2019, 32.
>
> [4] Wang Z, Li X, Wang J, et al. Learning Cut Selection for Mixed-Integer Linear Programming via Hierarchical Sequence Model[C]//The Eleventh International Conference on Learning Representations.
>
> ### **Response to the question 2:**
> To ensure fair comparisons, we tested a GNN algorithm based on imitation learning with heuristic pre-selection (GNN+BestEstimate) to eliminate the impact of "BestEstimate" on the solving results. The results are included in Appendix E.8 of the revised manuscript. We found that the filtered results were consistent with the original ones because nodes ranked lower by the heuristic "BestEstimate" were seldom chosen by the GNN (see the response to weakness 2).
>
> MaxSAT($n\in[80,100]$)
> | Method | time(s)  |nodes |
> |-------|-----|------|
> | SCIP      | 8.15  ± 1.64 |147.51  ± 2.02|
> | GNN+BestEstimate |  7.71  ± 1.93 |230.72  ± 3.30  |
> | TRGNN(Ours) |  **6.38  ± 1.85** |**134.93  ± 2.83** |
>
> GISP($n\in[60,70]$)
> | Method | time(s)  |nodes |
> |-------|-----|------|
> | SCIP      |  3.12  ± 1.30|77.10  ± 3.79|
> | GNN+BestEstimate | 3.26  ± 1.35 | 96.99  ± 3.66  |
> | TRGNN(Ours) |  **2.88  ± 1.28** |**80.50  ± 3.39**  |

---

> > ### Author Response · Authors · 2024-11-27
> >
> > Dear Reviewer c3Ua,
> >
> > We hope this message finds you well. The author response period for this submission was extended until the end of December 2nd.
> >
> > We have not yet received feedback from you. Ensuring that our response effectively addresses your concerns is a priority for us. Therefore, might we inquire if you have any additional questions or concerns?
> >
> > We appreciate your time and dedication committed to evaluating our work.
> >
> > Best Regards,
> >
> > The Authors of Submission 3329

---

> > > ### Comment · Reviewer_c3Ua · 2024-12-03
> > >
> > > Thank you very much for your reply, most concerns have been addressed, but some issues mentioned earlier have not yet been responded to:
> > >
> > > 1. The author only analyzed the time required from graph construction to inference for bipartite and tripartite graphs based on the current existing scale. It seems that the tripartite graph takes longer, but would the time gap become larger as the problem scale expands? In this case, the advantage of the rich representation brought by the tripartite graph is not so obvious, because usually, we expect to obtain a near-optimal solution in a short time.
> > >
> > > 2. As the authors mentioned, why not choose newer baselines like HEM? It seems that the methods reported in the paper are relatively old, and the performance is not the most SOTA in node selection currently

---

> > > > ### Author Response · Authors · 2024-12-04
> > > > **Follow-up Response to Reviewer c3Ua**
> > > >
> > > > Thank you for taking the time to read our responses. We address your newly raised concerns in order.
> > > >
> > > > ## Response to the question 1
> > > > First, regarding your point that "the tripartite graph takes longer", this is not accurate. We analyzed the time complexity of graph construction for both tripartite and bipartite graphs and showed that the **tripartite graph actually reduces the time complexity** compared to the bipartite graph. This is because the bipartite graph needs to construct a separate graph for each candidate node, whereas our tripartite approach incorporates all candidate nodes into a single graph, **avoiding the redundant representation of shared variables and constraints**. Given that the complexity of |LN| is at most O(n) and |E| is at most O(n × m), the complexity for the bipartite approach is at most O(n^2 × m), while for the tripartite approach, it is at most O(n^2 + n × m), where n denotes the number of variables and m denotes the number of constraints.
> > > >
> > > > Therefore, in response to your concern about whether the tripartite approach's graph construction time would increase more than that of the bipartite approach **as the problem scale expands**, our tripartite method does not lead to such an issue. In fact, it is even **quicker** than the bipartite approach in terms of graph construction time.
> > > >
> > > > Moreover, we would like to add that in practice, the increase in graph construction time due to scaling up is **significantly less than the increase in total solving time**. Since MILP problems are NP-hard, their solving time typically grows exponentially with problem size. In contrast, the time complexity for both tripartite and bipartite graphs only grows **polynomially** with the size of variables and constraints.
> > > >
> > > > We recorded the graph construction time for the tripartite method (TRGNN) and bipartite method (BRGNN) on a **transfer** dataset, summarizing these along with the previously tested times on the test dataset in the table below. We observed that while **graph construction time** does not increase significantly with scale, **the total solving time** for MaxSAT on the transfer dataset is nearly 10 times that of the test dataset, and for GISP, it is over 100 times that of the test dataset.
> > > >
> > > >
> > > > Graph Construction Time(s)
> > > > | Method | MaxSAT (Test) |GISP (Test) | MaxSAT (Transfer)  |GISP (Transfer) |
> > > > |-------|-----|------|----|------|
> > > > | BRGNN | 0.01  | 0.00 |0.02  | 0.00 |
> > > > | TRGNN(Ours) |  0.01  | 0.00 |0.01  | 0.00 |
> > > >
> > > >
> > > > ## Response to the question 2
> > > >
> > > > **About HEM**: HEM [1] is not a baseline for node selection. It focuses on **cutting plane selection**. Cutting plane selection and node selection are distinct modules. Cutting plane selection involves determining the number of cuts to add and which specific cuts to introduce. Therefore, the primary challenge lies in **balancing the benefits against the costs of introducing these cuts** and understanding the **combined effect of multiple cuts** on the constraint space. In cutting plane selection, the reduction of the constraint space by the cuts is immediate.
> > > >
> > > > In contrast, in node selection, the contribution of each selected node to solving the problem is **not immediate**. We need to provide a more sufficient global tree representation to **estimate the potential contribution of each node** to the convergence of the bounds, while also balancing the additional overhead introduced by the node selection process, such as path switching costs. Cutting plane selection and node selection are not directly related, and comparing studies on these entirely different problems is unnecessary.
> > > >
> > > >
> > > > **About the baselines in our paper:** We would like to clarify that the GNN baseline [2] we used for comparison are the **best-performing** works we could find that specifically address the problem of **node selection**.
> > > >
> > > >
> > > >
> > > > **References:**
> > > >
> > > > [1] Wang Z, Li X, Wang J, et al. Learning Cut Selection for Mixed-Integer Linear Programming via Hierarchical Sequence Model[C]//The Eleventh International Conference on Learning Representations.
> > > >
> > > > [2] Labassi A G, Chételat D, Lodi A. Learning to compare nodes in branch and bound with graph neural networks[J]. Advances in neural information processing systems, 2022, 35: 32000-32010.

---

### Official Review · Reviewer_6B24 · 2024-11-09

**Soundness:** 3
**Presentation:** 2
**Contribution:** 3
**Rating:** 6
**Confidence:** 4

**Summary:**

In this paper, the authors propose solving the node selection problem using a novel tripartite graph representation combined with reinforcement learning and a Graph Neural Network model. Additionally, the authors have theoretically proven that the tripartite graph representation is sufficient in information theory. The authors used a MDP to model the node selection problem and the reward function consists of three parts, including the updates to the global gap, the potential of a node to lead to an optimal solution and the normalized reward for path-switching steps. The experimental results show that the proposed TRGNN outperforms existing baselines that use learning-based methods on six NP-hard MILP problem benchmarks. Moreover, the trained model demonstrates a certain level of generalization.

**Strengths:**

（1）The paper is well organized. First, the tripartite graph representation of the branch-and-bound tree is explained, and it is proven that this representation can capture sufficient information. Then, the overall algorithm framework is presented, and the components of the MDP are described.

（2）The method of converting the branch-and-bound tree into a tripartite graph presented in the paper is worth learning, especially since the authors prove that the information obtained through the GNN on this graph captures sufficient information contained in the tree.

（3）The experimental results are promising, demonstrating that the approach outperforms recent machine learning algorithms as well as the state-of-the-art best estimate node selection rule.

**Weaknesses:**

（1）The presentation in the paper is not clear enough and lacks some necessary definitions, such as what a leaf node vertex is. I struggled to understand what the LN set represents until I re-read the paper and carefully examined Figure 1.

（2）For the branch-and-bound tree conversion method, there is a lack of concrete examples. Although a simple conversion is provided in Figure 1, it lacks an explanation of how each step is derived, and there is no description of the features on the edges.

**Questions:**

（1）Please address the issues raised in the weaknesses section.

（2）The paper uses a heuristic algorithm to initially select a subset of nodes, and then employs the model to choose one node from this subset. How do the results of this approach compare to the experimental results of directly using the heuristic algorithm to select nodes without employing the model?

---

> ### Author Response · Authors · 2024-11-20
> **Response to Reviewer 6B24**
>
> Thanks for your valuable feedback and suggestions. We will address the issues pointed out in the weakness section in order.
>
> ### **Response to the weaknesses 1:**
> The **leaf node vertices** represent candidate leaf nodes for selection strategies within the branch-and-bound tree. Once a node is selected, the branch-and-bound algorithm splits it into two new leaf nodes by adding constraints of the form $\{x_i\leq z\}$ or $\{x_i\geq z\}$, where $x_i$ is the variable chosen for branching and $z\in \mathbb{Z}$. **The LN set** is constructed from these leaf node vertices. We included these descriptions in the revised manuscript(p5,l223).
> ### **Response to the weaknesses 2:**
> We added an explanation of the conversion on page 5, line 250 of the revised manuscript: The conversion begins with the root node, where variables and constraints are represented as "variable vertices" and "constraint vertices", respectively.
> As the branching process progresses, each step adds new constraints to the leaf nodes, encapsulated as "leaf node vertices" in the graph. These additional constraints differentiate each leaf node's MILP problem from the root node's problem. Edges connect these leaf node vertices to variable vertices, forming a path that represents the sequence of branching decisions made from the root node to each leaf node. For example, in Figure 1, the candidate leaf node in the lower left undergoes two branching steps, adding constraints $y\leq 1$ and $x\leq 1$, represented by edges $e_1$ and $e_3$, respectively.
> ### **Response to the question:**
> We added additional experiments to evaluate the effectiveness of directly using the heuristic "BestEstimate".
> To demonstrate the impact of the tripartite graph representation on solving efficiency compared to bipartite graphs, we conducted an ablation study. In this study, we used the BestEstimate heuristic to pre-filter candidate nodes and then represented each candidate node with a bipartite graph within our reinforcement learning framework (BRGNN). The results are as follows:
>
> MaxSAT($n\in[80,100]$)
> | Method | time(s)  |wins |nodes |
> |-------|-----|------|----|
> | SCIP      | 8.15  ± 1.64 |3/50 |147.51  ± 2.02|
> | BestEstimate | 7.98  ± 1.68|11/50  |177.64  ± 2.13|
> | BRGNN |  7.92  ± 1.67 |5/50 |200.68  ± 2.05  |
> | TRGNN(Ours) |  **6.38  ± 1.85** |**31/50**  |**134.93  ± 2.83**  |
>
> GISP($n\in[60,70]$)
> | Method | time(s) |wins  |nodes |
> |-------|-----|----|-----|
> | SCIP      |  3.35  ± 1.30| 6/50 |94.81  ± 3.67|
> | BestEstimate |  3.53  ± 1.30 |8/50  |102.99  ± 3.18|
> | BRGNN | 3.20  ± 1.32 | 17/50 |97.27  ± 3.59  |
> | TRGNN(Ours) |  **3.03  ± 1.27** |**19/50**  |**83.72  ± 3.51**  |
>
> Experiment shows that the "BestEstimate" method performs similarly to the default SCIP settings. Our tripartite graph approach (TRGNN) improves over SCIP and "BestEstimate". Notably, the tripartite graph method outperforms the bipartite method, achieving improvements of 19.4% on the MaxSAT dataset and 2.5% on the GISP dataset. We included the results in Appendix E.6 of the revised manuscript.

---

> > ### Author Response · Authors · 2024-11-27
> >
> > Dear Reviewer 6B24,
> >
> > We hope this message finds you well. The author response period for this submission was extended until the end of December 2nd.
> >
> > We have not yet received feedback from you. Ensuring that our response effectively addresses your concerns is a priority for us. Therefore, might we inquire if you have any additional questions or concerns?
> >
> > We appreciate your time and dedication committed to evaluating our work.
> >
> > Best Regards,
> >
> > The Authors of Submission 3329

---

> > ### Comment · Reviewer_6B24 · 2024-11-28
> >
> > Thanks to your reply. Since there is not enough research on BnB algorithm from this perspective combined with deep learning, and I am not familiar with this direction, I won't change my score. But I think it's a good piece of work.

---

### Meta-Review · Area_Chair_hJjQ · 2024-12-17

**Metareview:**

This paper addresses the node selection problem in the branch-and-bound method for MILP. It proposes the TRGNN approach, which utilizes a tripartite graph representation and reinforcement learning. Theoretically, the tripartite graph can sufficiently represent the tree. Empirically, TRGNN outperforms existing methods in solving MILPs and shows good generalization.

The strengths of the paper are its innovative approach, with a novel graph representation and effective use of reinforcement learning. The experimental results are also promising. However, it has weaknesses. The graph representation's complexity and computational cost need further analysis. There's a lack of comprehensive baseline comparisons and detailed hyperparameter analysis. The submission could also benefit from more discussion on scalability and practical implementation details. Overall, the paper's contribution is significant enough to warrant acceptance, given the authors' efforts to address concerns and the potential impact on the field.

**Additional Comments On Reviewer Discussion:**

During the rebuttal period, reviewers raised several points. These included issues like the clarity of presentation, the necessity and complexity of the tripartite graph representation, the choice of baselines, time complexity analysis, the impact of heuristic algorithms, and the generalization of the method. The authors addressed these by adding detailed explanations, conducting ablation studies, providing more experimental results, clarifying the role and performance of heuristic algorithms, and discussing potential improvements and limitations. In weighing these points, the authors' responsiveness and the substantial contribution of the proposed approach despite the areas for improvement led to the decision to accept the paper.

---

### Decision · Program_Chairs · 2025-01-22

Accept (Poster)